# Alignment Between the Decision-Making Logic of LLMs and Human Cognition: A Case Study on Legal LLMs

## Abstract

This paper presents a method to evaluate the alignment between the decision-making logic of Large Language Models (LLMs) and human cognition in a case study on legal LLMs. Unlike traditional evaluations on language generation results, we propose to evaluate the correctness of the detailed decision-making logic of an LLM behind its seemingly correct outputs, which represents the core challenge for an LLM to earn human trust. To this end, we quantify the interactions encoded by the LLM as primitive decision-making logic, because recent theoretical achievements (Li & Zhang, 2023; Ren et al., 2024) have proven several mathematical guarantees of the faithfulness of the interaction-based explanation. We design a set of metrics to evaluate the detailed decision-making logic of LLMs. Experiments show that even when the language generation results appear correct, a significant portion of the internal inference logic contains notable issues[1].

## 1 Introduction

The trustworthiness and safety of Large Language Models (LLMs) present significant challenges for their deployment in high-stake tasks (OpenAI, 2023; Wei et al., 2023). Previous evaluation methods mainly evaluated the correctness of language generation results, in terms of value alignment and hallucination problems (Bang et al., 2023; Ji et al., 2023b;a; Shen et al., 2023).

In this study, we hope to go beyond the long-tail evaluation of the generation results, and focus on *the correctness of the detailed decision-making logic used by the LLM behind the language generation result*. We focus on the legal LLM as a case study, and the legal LLM may use significantly incorrect information to make judgment, even when the generation result is correct. The alignment of decision-making logic between the AI model and human cognition is crucial for alleviating the common fear of AI models. The alignment of internal logic via communication is the reason why people naturally trust each other. Particularly, in high-stakes tasks such as autonomous driving (Grigorescu et al., 2020), the lack of alignment between AI models and human users makes people would rather delegate work to humans and tolerate potential errors, than trust highly accurate AI models.

Therefore, this paper aims to explore the possibility of aligning the decision-making logic for the confidence score of the LLM's judgment with human cognition. To this end, exploring the mathematical feasibility of faithfully explaining the output score of a neural network as a few interpretable logical patterns has become a new emerging theoretical problem in explainable AI, and about 20 papers have been published in three years (see related work in Appendix A). Typically, Li & Zhang (2023); Ren et al. (2024) have proved the *universal-matching property* and *sparsity propoerty*, and mathematically guaranteed that a DNN usually only encodes a small number of interactions between input variables, and these interactions act as **primitive decision-making logic**, which well predicts the confidence of the network prediction on various input variations.

As Figure 1 shows, an *interaction* measures the nonlinear relationship between input tokens of an input legal case encoded by the LLM. For instance, given an input sentence such as "*Andy threatened Bob and took his smartphone,*" the LLM may trigger an interaction between a set of input tokens $S =$

---

[1]The names used in the legal cases follow an alphabetical convention, *e.g.*, Andy, Bob, Charlie, etc., which do not represent any bias against actual individuals.

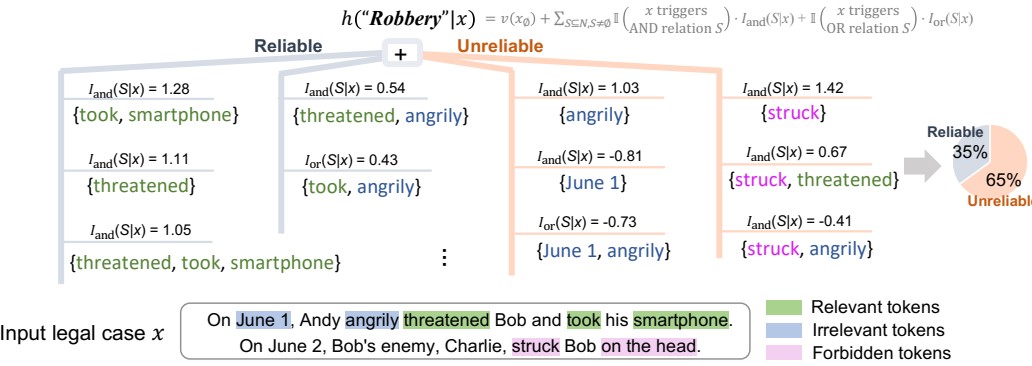

Figure 1: AND-OR interactions that explain the decision-making logic of a legal LLM. The surrogate logical model well estimates the confidence of the LLM making the judgment "***Robbery***" for Andy, $h(\text{"Robbery"}|\mathbf{x}) = v(\text{"Robbery"}|\mathbf{x})$, no matter how we randomly mask the input $\mathbf{x}$.

$\{threatened, took, smartphone\} \subseteq N$, and the interaction makes a numerical effect $I(S)$ that boosts the confidence of inferring the judgment of "*robbery*." Besides, Zhou et al. (2024) demonstrated that the complexity of interactions directly determined the generalization power of a DNN.

Despite above achievements, previous studies have pointed out that the next breakthrough point is to examine the correctness of the detailed decision-making logic used by the LLM, which have not been explored yet (Deng et al., 2024b; Li & Zhang, 2023; Cheng et al., 2024; Ren et al., 2024; Chen et al., 2024; Zhou et al., 2024).

In this paper, we extract all interactions that determine a legal LLM's confidence score of the true judgment, and we evaluate the alignment between the extracted interactions and human cognition of the legal case. To this end, we categorize all input tokens involved in the interactions into three types, *i.e.*, the *relevant*, *irrelevant*, and *forbidden*[2] tokens, based on the ground-truth relevance to the judgment. This enables us to distinguish *reliable interaction effects* and *unreliable interaction effects*. For example, as Figure 1 shows, the legal LLM makes the judgment of "*robbery*" on Andy who takes Bob's smartphone under threat. In this way, AND interactions involving "*threatened,*" "*took,*" and "*smartphone*" are supposed to be the correct reason for the judgment, thereby being identified as *reliable interaction effects*. In comparison, the OR interaction between "*June 1*" and "*angrily*" incorrectly attributes the judgment of "*robbery*" to the unreliable sentimental token "*angrily*." It is because we should use the real action "*threatened*" to make the judgment, rather than the sentimental token "*angrily*." The unreliable interaction also includes the AND interaction between "*struck*" and "*threatened,*" which incorrectly attributes the judgment on Andy to the forbidden token "*struck,*" *i.e.*, an action **not** taken by Andy.

In this way, we design new metrics based on these interactions to quantify the ratio of reliable interaction effects and that of unreliable ones used by the LLM to generate the target judgement, so as to evaluate the alignment between the LLM's logic and human cognition.

The contributions of this paper can be summarized as follows. We propose to utilize interaction-based explanations to evaluate the correctness of decision-making logic encoded by a LLM. We design new metrics to quantify reliable and unreliable interaction effects *w.r.t.* their alignment with human cognition of the judgment. Experiments on both English legal LLM and Chinese legal LLM show that both LLMs used a significant number of incorrect interactions for inference, although these LLMs all exhibited high accuracy in judgment prediction.

## 2 ALIGNMENT BETWEEN THE LLM AND HUMAN COGNITION

### 2.1 PRELIMIARIES: INTERACTIONS

Although there is no widely-accepted definition of concepts, which is an interdisciplinary issue across cognitive science, neuroscience, artificial intelligence, and mathematics, the theory of interactions has shown promise in explaining the primitive inference patterns encoded by the DNN. A

---

[2]The forbidden tokens are usually informative tokens but should not be used for judgments, *e.g.*, tokens of criminal actions that are **not** taken by the defendant.

series of properties (Li & Zhang, 2023; Ren et al., 2023a; 2024) have been proposed as mathematical guarantees for the faithfulness of the interaction-based explanations.

**Definition of AND-OR interactions.** Given an input sample $\mathbf{x} = [x_1, x_2, \cdots, x_n]^\intercal$ with $n$ input variables indexed by $N = \{1, 2, ..., n\}$, where each input variable can represent a token, a word, or a phrase/short sentence. Then, let $v(\mathbf{x}) \in \mathbb{R}$ denote the *scalar* confidence of generating the target output. For example, the target output can be set to a sequence of $m$ ground-truth tokens $[y_1, y_2, \cdots, y_m]$ generated by the LLM. In this way, the scalar confidence of language generation $v(\mathbf{x})$ can be defined as follows.

$$v(\mathbf{x}) \stackrel{\text{def}}{=} \sum_{t=1}^{T} \log \frac{p(y = y_t | \mathbf{x}, \mathbf{Y}_t^{\text{previous}})}{1 - p(y = y_t | \mathbf{x}, \mathbf{Y}_t^{\text{previous}})} \tag{1}$$

where $\mathbf{Y}_t^{\text{previous}} \stackrel{\text{def}}{=} [y_1, y_2, \cdots, y_{t-1}]^\intercal$ represents the sequence of the previous $(t-1)$ tokens before generating the $t$-th token. $p(y = y_t | \mathbf{x}, \mathbf{Y}_t^{\text{previous}})$ denotes the probability of generating the $t$-th token, given the input sentence $\mathbf{x}$ and the previous $(t-1)$ tokens. In particular, $\mathbf{Y}_1^{\text{previous}} = []$.

To explain the inference patterns behind the confidence score $v(\mathbf{x})$, Ren et al. (2024); Shen et al. (2023) show that an LLM usually encodes a set of interactions between input variables (tokens or phrases) to compute $v(\mathbf{x})$. There are two types of interactions, *i.e.*, the AND interaction and the OR interaction. Each AND interaction and each OR interaction *w.r.t.* $S \subseteq N, S \neq \emptyset$ have specific numerical effects $I_{\text{and}}(S|\mathbf{x})$ and $I_{\text{or}}(S|\mathbf{x})$ to the network output, respectively, which are computed as follows.

$$I_{\text{and}}(S|\mathbf{x}) \stackrel{\text{def}}{=} \sum_{T \subseteq S} (-1)^{|S|-|T|} v_{\text{and}}(\mathbf{x}_T), \quad I_{\text{or}}(S|\mathbf{x}) \stackrel{\text{def}}{=} -\sum_{T \subseteq S} (-1)^{|S|-|T|} v_{\text{or}}(\mathbf{x}_{N \setminus T}) \tag{2}$$

where $\mathbf{x}_T$ denotes the masked sample[3], where all embeddings of input variables in $N \setminus T$ are masked. $v(\mathbf{x}_T) \in \mathbb{R}$ denotes the confidence score of generating the $m$ tokens $[y_1, y_2, \cdots, y_m]$ given the masked sample $\mathbf{x}_T$. $v(\mathbf{x}_T)$ is decomposed into the component for AND interactions $v_{\text{and}}(\mathbf{x}_T) = 0.5v(\mathbf{x}_T) + \gamma_T$ and the component for OR interactions $v_{\text{or}}(\mathbf{x}_T) = 0.5v(\mathbf{x}_T) - \gamma_T$, subject to $v_{\text{and}}(\mathbf{x}_T) + v_{\text{or}}(\mathbf{x}_T) = v(\mathbf{x}_T)$.

**Extracting AND-OR interactions.** According to Equation (2), the extraction of interactions is implemented by learning parameters $\{\gamma_T\}$. We follow (Zhou et al., 2024) to learn parameters $\{\gamma_T | T \subseteq N\}$, and extract the sparest (the simplest) AND-OR interaction explanation via the LASSO-like loss, *i.e.*, $\min_{\{\gamma_T\}} \sum_{S \subseteq N, S \neq \emptyset} [|I_{\text{and}}(S|\mathbf{x})| + |I_{\text{or}}(S|\mathbf{x})|]$. In this way, we exhaustively compute interaction effects $I_{\text{and}}(S|\mathbf{x})$ and $I_{\text{or}}(S|\mathbf{x})$ for all $(2^n - 1)$ non-empty combinations $\emptyset \neq S \subseteq N$. **Ren et al. (2024) have proven that most interactions have almost zero effects $I_{\text{and/or}}(S|\mathbf{x})$, and an LLM usually activates only 100-200 AND-OR interactions with salient effects. These salient interactions are taken as the AND-OR logic really encoded by the LLM.**

Algorithm 1 in the appendix shows the pseudo-code of extracting AND-OR interactions.

**Why do AND-OR interactions faithfully explain the logic encoded by the LLM?** Lots of theoretical achievements ranging from (Harsanyi, 1963) to (Li & Zhang, 2023; Ren et al., 2023a; 2024) have proven several properties to guarantee that **the AND-OR interactions faithfully represent the AND-OR logic encoded by the LLM.** According to Theorem 1, let $h(\cdot)$ denote a surrogate logical model constructed based on non-zero interactions. As Figure 6 shows, it is proven that this surrogate logical model $h(\cdot)$ can accurately fit the confidence scores of the LLM $v(\cdot)$ on all $2^n$ masked samples $\{\mathbf{x}_T | T \subseteq N\}$, *i.e.*, $\forall T \subseteq N, v(\mathbf{x}_T) = h(\mathbf{x}_T)$, no matter how we randomly mask the input sample $\mathbf{x}$ in $2^n$ different masking states $T \subseteq N$. This property is termed *universal-matching property*.

**Theorem 1 (Universal matching property, proof in Appendix B)** *Given an input sample $\mathbf{x}$, the network output score $v(\mathbf{x}_T) \in \mathbb{R}$ on each masked sample $\{\mathbf{x}_T | T \subseteq N\}$ can be well matched by a surrogate logical model $h(\mathbf{x}_T)$ on each masked sample $\{\mathbf{x}_T | T \subseteq N\}$. The surrogate logical model $h(\mathbf{x}_T)$ uses the sum of AND interactions and OR interactions to accurately fit the network output*

---

[3]To obtain the masked sample $\mathbf{x}_T$, we mask the embedding of each input variable $i \in N \setminus T$ with the baseline value $b_i$ to represent its masked state. Please see Appendix G.4 for details.

*score $v(\mathbf{x}_T)$.*

$$\forall T \subseteq N, v(\mathbf{x}_T) = h(\mathbf{x}_T).$$

$$h(\mathbf{x}_T) = v(\mathbf{x}_\emptyset) + \sum_{S \subseteq N, S \neq \emptyset} \mathbb{1}(\begin{array}{c}\mathbf{x}_T \text{ triggers} \\ \text{AND relation } S\end{array}) \cdot I_{\text{and}}(S|\mathbf{x}_T) + \mathbb{1}(\begin{array}{c}\mathbf{x}_T \text{ triggers} \\ \text{OR relation } S\end{array}) \cdot I_{\text{or}}(S|\mathbf{x}_T)$$

$$= \underbrace{v(\mathbf{x}_\emptyset) + \sum_{S \subseteq T, S \neq \emptyset} I_{\text{and}}(S|\mathbf{x}_T)}_{v_{\text{and}}(\mathbf{x}_T)} + \underbrace{\sum_{S \subseteq N, S \cap T \neq \emptyset} I_{\text{or}}(S|\mathbf{x}_T)}_{v_{\text{or}}(\mathbf{x}_T)} \quad (3)$$

Specifically, each non-zero AND interaction $I_{\text{and}}(S|\mathbf{x})$ represents the AND relationship between all variables in $S$. For instance, consider an input sentence "*the company is a legal person*" in a language generation task. The co-appearance of two words $S = \{legal, person\} \subseteq N$ forms a specialized legal concept and contributes a numerical effect $I_{\text{and}}(S|\mathbf{x})$ to push the LLM's output *w.r.t.* the legal entity. Exclusively inputting either word in $S$ will not make such an effect.

Analogously, each non-zero OR interaction $I_{\text{or}}(S|\mathbf{x})$ indicates the OR relationship between all variables in $S$. For example, let us consider an input sentence "*he robbed and assaulted a passerby*". The presence of either word in $S = \{robbed, assaulted\}$ activates the OR relationship and contributes an effect $I_{\text{or}}(S|\mathbf{x})$ to push the LLM towards a guilty verdict.

Besides the *universal-matching property* in Theorem 1, the *sparsity property* of interactions is also proven (Ren et al., 2024). *I.e.,* most AND-OR interactions have almost zero effects, *i.e.,* $I(S|\mathbf{x}) \approx 0$, which can be regarded as negligible noise patterns. Only a small set of interactions, denoted by $\Omega = \{S \subseteq N : |I(S|\mathbf{x})| > \tau\}$, where $\tau$ is a scalar threshold, have considerable effects. Therefore, Lemma 1 shows that the surrogate logical model $h(\cdot)$ on all $2^n$ masked samples $\{\mathbf{x}_T | T \subseteq N\}$ usually can be approximated by a small set of salient AND interactions $\Omega^{\text{and}}$ and salient OR interactions $\Omega^{\text{or}}$, *s.t.,* $|\Omega^{\text{and}}|, |\Omega^{\text{or}}| \ll 2^n$.

**Lemma 1 (Sparsity property, proof in Appendix C)** *The surrogate logical model $h(\mathbf{x}_T)$ on each randomly masked sample $\mathbf{x}_T, T \subseteq N$ mainly uses the sum of a small number of salient AND interactions and salient OR interactions to approximate the network output score $v(\mathbf{x}_T)$.*

$$v(\mathbf{x}_T) = h(\mathbf{x}_T) \approx v(\mathbf{x}_\emptyset) + \sum_{S \in \Omega^{\text{and}}} \mathbb{1}(\begin{array}{c}\mathbf{x}_T \text{ triggers} \\ \text{AND relation } S\end{array}) \cdot I_{\text{and}}(S|\mathbf{x}_T) + \sum_{S \in \Omega^{\text{or}}} \mathbb{1}(\begin{array}{c}\mathbf{x}_T \text{ triggers} \\ \text{OR relation } S\end{array}) \cdot I_{\text{or}}(S|\mathbf{x}_T) \quad (4)$$

**The above *universal-matching property* and *sparsity property* theoretically guarantee the faithfulness of the interaction-based explanation.**

## 2.2 RELEVANT TOKENS, IRRELEVANT TOKENS, AND FORBIDDEN TOKENS

According to above achievements, we can take a small set of salient AND-OR interactions as the faithful explanation for the decision-making logic used by the legal LLM. Thus, in this subsection, we annotate the *relevant, irrelevant,* and *forbidden* tokens in the input legal case, in order to accurately identify the reliable and unreliable interactions encoded by the LLM (see Figure 1). Specifically, the set of all input variables $N$ is partitioned into three mutually disjoint subsets, *i.e.,* the set of relevant tokens $\mathcal{R}$, the set of irrelevant tokens $\mathcal{I}$, and the set of forbidden tokens $\mathcal{F}$, subject to $\mathcal{R} \cup \mathcal{I} \cup \mathcal{F} = N$, with $\mathcal{R} \cap \mathcal{I} = \emptyset$, $\mathcal{R} \cap \mathcal{F} = \emptyset$, and $\mathcal{I} \cap \mathcal{F} = \emptyset$, according to human cognition.

*Relevant tokens* refer to tokens that are closely related to or serve as the direct reason for the judgment, according to human cognition. For instance, given an input legal case "*on June 1, during a conflict on the street, Andy stabbed Bob with a knife, causing Bob's death,*"[1] the legal LLM provides judgment "*murder*" for Andy. In this case, the input variables can be set as $N = \{[on\ June\ 1], [during\ a\ conflict], [on\ the\ street], [Andy\ stabbed\ Bob\ with\ a\ knife], [causing\ Bob's\ death]\}$. $\mathcal{R} = \{[Andy\ stabbed\ Bob\ with\ a\ knife], [causing\ Bob's\ death]\}$ are the direct reason for the judgment, thereby being annotated as *relevant tokens*, where all tokens in the brackets [] are taken as a single input variable.

*Irrelevant tokens* refer to tokens that are not strongly related to or are not the direct reason for the judgment, according to human cognition. For instance, in the above input legal case, the set of irrelevant tokens are annotated as $\mathcal{I} = \{[on\ June\ 1], [during\ a\ conflict], [on\ the\ street]\}$. For example,

the input variable like "*during a conflict*" may influence Andy's behavior "*Andy stabbed Bob with a knife*," but it is the input variable "*Andy stabbed Bob with a knife*" that directly contributes to the legal judgment of "*murder*," rather than the input variable "*during a conflict.*"

*Forbidden tokens* are usually common tokens widely used in legal cases, but the use of forbidden tokens may lead to significant incorrect logic. For instance, in a legal case involving multiple individuals, such as "*Andy assaulted Bob on the head, causing minor injuries. Charlie stabbed*

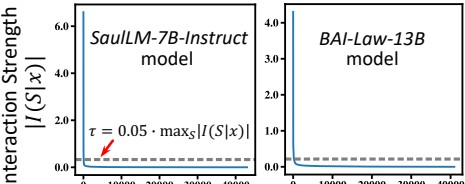

Figure 2: Sparsity of interactions. We show the strength of different AND-OR interactions $|I(S|\mathbf{x})|$ extracted from different samples in a descending order. Only about 0.5% interactions had salient effects.

*Bob with a knife, causing Bob's death,*"[1] the legal LLM assigns the judgment of "*assault*" to Andy. Let the set of all input variables be $N = \{$[*Andy assaulted Bob on the head*], [*causing minor injuries*], [*Charlie stabbed Bob with a knife* ], [*causing Bob's death*]$\}$. Although the input variables "*Charlie stabbed Bob with a knife*" and "*causing Bob's death*" are naturally all represent crucial facts for judgement, they should not influence the judgment for Andy, because these words describe the actions of Charlie, not actions of Andy. Therefore, these input variables are categorized as forbidden tokens, $\mathcal{F} = \{$[*Charlie stabbed Bob with a knife*], [*causing Bob's death*]$\}$.

### 2.3 Reliable and unreliable interaction effects

The categorization of *relevant*, *irrelevant*, and *forbidden* tokens enables us to disentangle the reliable and unreliable decision-making logic used by a legal LLM. As introduced in Section 2.1, we use interactions as the decision-making logic encoded by a legal LLM. Thus, in this subsection, we decompose the overall interaction effects in Equation (2) into reliable and unreliable interaction effects. *Reliable interaction effects* are interaction effects that align with human cognition, which usually contain relevant tokens and exclude forbidden tokens. In contrast, *unreliable interaction effects* are interaction effects that do not match human cognition, which are attributed to irrelevant or forbidden tokens.

**Visualization of AND-OR interactions.** Before defining reliable and unreliable interaction effects, let us first visualize the AND-OR interactions extracted from two legal LLMs, SaulLM-7B-Instruct (Colombo et al., 2024) and BAI-Law-13B (Institute, 2023). SaulLM-7B-Instruct was an English legal LLM, trained on a corpus of over 30 billion English legal tokens. BAI-Law-13B was a Chinese legal LLM, fine-tuned on Chinese legal corpora. We evaluated the legal LLMs on the CAIL2018 dataset (Xiao et al., 2018)[5], just like how (Feng et al., 2022; Fei et al., 2023) did. Figure 2 shows the sparsity of interactions extracted from the legal LLMs. Interaction strength $|I(S|\mathbf{x})|$ of all AND-OR interactions extracted from all legal cases were shown in a descending order. We found that most of the interactions had negligible effect.

Figure 1 further provides an example of using AND-OR interactions to explain the decision-making logic of a legal LLM. The legal LLM correctly attributes the judgment of "*robbery*" to interactions involving the tokens "*took*," "*smartphone*," and "*threatened.*" However, the legal LLM also uses the irrelevant tokens ("*angrily*" and "*June 1*"), and the forbidden tokens ("*struck*" and "*on the head*") to compute the confidence score of the judgment of "*robbery*," which obviously represents incorrect decision-making logic.

In this way, we define *reliable* and *unreliable* interaction effects for AND and OR interactions, respectively, as follows.

**For AND interactions.** Because the AND interaction $I_{\text{and}}(S|\mathbf{x})$ is activated only when all input variables (tokens or phrases) in $S$ are present in the input legal case, the reliable interaction effect for AND interaction $I_{\text{and}}^{\text{reliable}}(S|\mathbf{x})$ *w.r.t.* $S$ must include relevant tokens in $\mathcal{R}$, *i.e.*, $S \cap \mathcal{R} \neq \emptyset$, and completely exclude forbidden tokens in $\mathcal{F}$, *i.e.*, $S \cap \mathcal{F} = \emptyset$. Otherwise, if $S$ contains any forbidden tokens in $\mathcal{F}$, or if $S$ does not contains any relevant tokens in $\mathcal{R}$, then the AND interaction $I_{\text{and}}(S|\mathbf{x})$ represents an incorrect logic for judgment. In this way, the reliable and unreliable AND interaction

effects *w.r.t.* $S$ can be computed as follows.

$$\text{if} \quad S \cap \mathcal{F} = \emptyset, S \cap \mathcal{R} \neq \emptyset \quad \text{then} \quad I_{\text{and}}^{\text{reliable}}(S|\mathbf{x}) = I_{\text{and}}(S|\mathbf{x}), \quad I_{\text{and}}^{\text{unreliable}}(S|\mathbf{x}) = 0$$
$$\text{otherwise}, \quad I_{\text{and}}^{\text{reliable}}(S|\mathbf{x}) = 0, \quad I_{\text{and}}^{\text{unreliable}}(S|\mathbf{x}) = I_{\text{and}}(S|\mathbf{x}) \quad (5)$$

**For OR interactions.** The OR interaction $I_{\text{or}}(S|\mathbf{x})$ affects the LLM's output when any input variable (token or phrase) in $S$ appears in the input legal case. Therefore, we can define the reliable effect $I_{\text{or}}^{\text{reliable}}(S|\mathbf{x})$ as the numerical component in $I_{\text{or}}(S|\mathbf{x})$ allocated to relevant input variables in $S \cap \mathcal{R}$. To this end, just like in (Deng et al., 2024b), we uniformly allocate the OR interaction effects to all input variables in $S$. The reliable and unreliable interactions effects are those allocated to relevant variables, and those allocated to irrelevant and forbidden variables, respectively.

$$\forall S \subseteq N, S \neq \emptyset, I_{\text{or}}^{\text{reliable}}(S|\mathbf{x}) = \frac{|S \cap \mathcal{R}|}{|S|} \cdot I_{\text{or}}(S|\mathbf{x}), I_{\text{or}}^{\text{unreliable}}(S|\mathbf{x}) = \left(1 - \frac{|S \cap \mathcal{R}|}{|S|}\right) \cdot I_{\text{or}}(S|\mathbf{x}) \quad (6)$$

## 2.4 EVALUATION METRICS

In this subsection, we design a set of metrics to evaluate the alignment quality between the interactions encoded by the LLM and human cognition.

**Ratio of reliable interaction effects.** Definition 1 introduces the ratio of reliable interaction effects that align with human cognition to all salient interaction effects. Here, we focus on the small number of salient interactions in $\Omega^{\text{and}}$ and $\Omega^{\text{or}}$, rather than conduct evaluation on interactions effects of all $2^n$ subsets $S \subseteq N$. This is because salient interactions can be taken as primitive decision-making logic of an LLM, while all other interactions have negligible effects and represent noise patterns.

**Definition 1 (Ratio of reliable interaction effects)** *Given an LLM, the ratio of reliable interaction effects to all salient interaction effects $s^{\text{reliable}}$ is computed as follows.*

$$s^{\text{reliable}} = \frac{\sum_{\Omega^{\text{and}}} |I_{\text{and}}^{\text{reliable}}(S|\mathbf{x})| + \sum_{\Omega^{\text{or}}} |I_{\text{or}}^{\text{reliable}}(S|\mathbf{x})|}{\sum_{\Omega^{\text{and}}} |I_{\text{and}}(S|\mathbf{x})| + \sum_{\Omega^{\text{or}}} |I_{\text{or}}(S|\mathbf{x})|} \quad (7)$$

A larger value of $s^{\text{reliable}} \in [0, 1]$ indicates that a higher proportion of interaction effects align with human cognition.

**Interaction distribution over different orders.** Zhou et al. (2024) have found that the low-order interactions usually exhibit stronger generalization power[4] than high-order interactions. *I.e.*, low-order interactions learned from training samples are more likely to be transferred to (appear in) testing samples. Please see Appendix E for the definition and quantification of the generalization power of interactions over different orders. Specifically, the *order* is defined as the number of input variables in $S$, *i.e.*, order$(S) = |S|$. In general, high-order interactions (complex interactions) between a large number of input variables are usually less generalizable[4] than low-order (simple) interactions.

Therefore, we utilize the distribution of interactions over different orders as another metric, which evaluates the generalization power of the decision-making logic used by the LLM. Specifically, we use $Salient^+(o) = \sum_{\text{op} \in \{\text{and,or}\}} \sum_{S \in \Omega^{\text{op}}, |S|=o} \max(0, I_{\text{op}}(S|\mathbf{x}))$ to quantify the overall strength of positive salient interactions, and use $Salient^-(o) = \sum_{\text{op} \in \{\text{and,or}\}} \sum_{S \in \Omega^{\text{op}}, |S|=o} \min(0, I_{\text{op}}(S|\mathbf{x}))$ to quantify the overall strength of negative salient interactions. A well-trained legal LLM tends to model low-order interactions, while an over-fitted LLM (potentially due to insufficient data or inadequate data cleaning) usually relies more on high-order interactions.

**Ratio of reliable interaction effects of each order.** We categorize all salient interaction effects by their orders, so that for all salient interactions of each $o$-th order, we can compute the ratio of reliable interaction effects.

---

[4]The generalization power of an interaction is defined as the transferability of this interaction from training samples to test samples. Specifically, if an interaction pattern $S \subseteq N$ frequently occurs in the training set, but rarely appears in the test set, then the interaction pattern $S$ exhibits low generalization power. Conversely, if an interaction pattern $S$ consistently appears in both the training and test sets, it demonstrates high generalization power. Please see Appendix E for details.

**Definition 2 (Ratio of reliable interaction effects of each order)** *The ratio of reliable interaction effects to all positive salient interaction effects of the o-th order is measured by* $s_o^{\text{reliable},+} = \frac{Reliable^+(o)}{Salient^+(o)+\epsilon}$. *Similarly, the ratio of reliable interaction effects to all negative salient interaction effects of the o-th order is measured by* $s_o^{\text{reliable},-} = \frac{|Reliable^-(o)|}{|Salient^-(o)|+\epsilon}$. *$Reliable^+(o) = \sum_{\text{op}\in\{\text{and,or}\}} \sum_{S\in\Omega^{\text{op}},|S|=o} \max(0, I_{\text{op}}^{\text{reliable}}(S|\mathbf{x}))$ represents the overall strength of positive reliable interactions of the o-th order, and $Reliable^-(o) = \sum_{\text{op}\in\{\text{and,or}\}} \sum_{S\in\Omega^{\text{op}},|S|=o} \min(0, I_{\text{op}}^{\text{reliable}}(S|\mathbf{x}))$ represents the overall strength of negative reliable interactions of the o-th order. $\epsilon$ is a small constant to avoid dividing 0.*

According to the findings in (Zhou et al., 2024), low-order interactions generally represent stable patterns that are frequently used across a large number of legal cases. Thus, if a considerable ratio of low-order interactions contain unreliable effects, it suggests that training data may have a clear bias, which makes the LLM stably learns unreliable interactions. In comparison, since high-order interactions typically exhibit poor generalization power, unreliable effects in high-order interactions are usually attributed to the memorization of hard/outlier samples. Consequently, low-order unreliable interactions are are mainly owing to stable bias in the training data, while high-order unreliable interactions often indicates that the LLM learns outlier features.

## 3 EXPERIMENT

In this section, we conducted experiments to evaluate the alignment quality between the decision-making logic of the legal LLM and human cognition. In this way, we identified potential representation flaws behind the seemingly correct language generation results of legal LLMs.

We applied two off-the-shelf legal LLMs, SaulLM-7B-Instruct (Colombo et al., 2024) and BAI-Law-13B (Institute, 2023), which were trained for legal judgment prediction on English legal corpora and Chinese legal corpora, respectively. Appendix F shows the accuracy of these LLMs. Given an input legal case, the LLM predicted the judgment result based on the fact descriptions of the legal case. We explained judgments made on legal cases in the CAIL2018 dataset (Xiao et al., 2018), which contained 2.6 million Chinese legal cases, for both legal LLMs[5]. Figure 6 shows the universal-matching property of the extracted interactions, *i.e.*, when we randomly masked input variables in the legal case, we could always use the interactions to accurately match the real confidence scores of the judgment estimated by the LLM.

To simplify the explanation and avoid ambiguity, we only explained the decision-making logic on legal cases, which were correctly judged by the LLM. For each input legal case, we manually selected some informative tokens or phrases as input variables. Some tokens or phrases were annotated as relevant tokens in $\mathcal{R}$, while others were identified as irrelevant tokens in $\mathcal{I}$. It was ensured that the removal of all input variables would substantially change the legal judgment result.

We extracted AND-OR interactions that determined the confidence score $v(\mathbf{x})$ of generating judgment results with a sequence of tokens, according to Equation (1). To accurately identify and analyze potential representation flaws from these interactions, in this paper, we mainly focused on potential representation flaws *w.r.t.* legal judgments in the following three types, *i.e.*, (1) judgments influenced by unreliable sentimental tokens, (2) judgments affected by incorrect entity matching, and (3) judgments biased by discrimination in occupation.

**Problem 1: making judgments based on unreliable sentimental tokens.** We observed that although legal LLMs achieved relatively high accuracy in predicting judgment results (see Appendix F), a considerable number of interactions contributing to the confidence score $v(\mathbf{x})$ were attributed to semantically irrelevant or unreliable sentimental tokens. The legal LLM was supposed to focus more on real criminal actions, than unreliable sentimental tokens behind the actions, when criminal actions had been given. We believed these indicated potential representation flaws behind

---

[5]To ensure a fair comparison, we conducted experiments using the same dataset across both legal LLMs. For the BAI-Law-13B model, which was a Chinese legal LLM, we directly analyzed the Chinese legal cases from the CAIL2018 dataset. For the SaulLM-7B-Instruct model, which was an English legal LLM, we translated these Chinese legal cases into English and performed the analysis on the translated cases, to enable fair comparisons. Please see Appendix G.5 for details.

the seemingly correct legal judgments produced by legal LLMs. To evaluate the impact of unreliable sentimental tokens on both the SaulLM-7B-Instruct and BAI-Law-13B models, we annotated tokens that served as the direct reason for the judgment as relevant tokens in $\mathcal{R}$, and those that were not the direct reason for the judgment as irrelevant tokens in $\mathcal{I}$, *e.g.*, semantically irrelevant tokens and unreliable sentimental tokens behind real criminal actions.

Figure 3 shows the legal case, which showed *Andy had a conflict with Bob and attacked Bob, committing an assault*. In this case, tokens like *"began to,"* *"causing,"* and sentiment-driven tokens such as *"dissatisfaction"* in $\mathcal{I}$ were irrelevant to the judgment result, according to human cognition, because unreliable sentimental tokens only served as explanations for criminal actions. Thus, once an actual action had been taken, the unreliable sentimental tokens were supposed to make minimal conditional contributions to the legal judgment result. The judgment should be based exclusively on tokens such as *"fight chaotically,"* *"threw a punch,"* and *"fall into a coma,"* which were annotated as relevant tokens in $\mathcal{R}$. We found that some decision-making logic encoded by the SaulLM-7B-Instruct model aligned well with human cognition, *i.e.*, identifying reliable interactions containing relevant tokens as the most salient interactions. However, this model also modeled lots of unreliable interactions as salient interactions, such as interactions containing irrelevant tokens *"dissatisfaction"* and *"anger,"* which revealed potential flaws in its decision-making logic.

In comparison, we evaluated the above legal case on the BAI-Law-13B model, as shown in Figure 3. The SaulLM-7B-Instruct model exhibited a reliable interaction ratio of $s^{\text{reliable}} = 71.5\%$, while the BAI-Law-13B model encoded a lower ratio of reliable interaction effects, $s^{\text{reliable}} = 61.2\%$. The BAI-Law-13B model encoded about 10% less reliable interactions, and used $s^{\text{unreliable}} = 1 - s^{\text{reliable}} = 38.8\%$ unreliable interaction effects to compute the confidence score $v(\mathbf{x})$. For example, reliable interactions encoded by the BAI-Law-13B model included the AND interaction $S = \{\text{"threw a punch"}\}$, which contributed the highest interaction effect 0.34. The unreliable interactions included the AND interaction $S = \{\text{"anger"}\}$, which contributed 0.03. The unreliable sentimental token should not be used to determine the judgment, when the action *"threw a punch"* caused by *"anger"* had been given as a more direct reason. Additional examples of making judgments based on unreliable sentimental tokens are provided in Appendix G.1.

**Problem 2: making judgments based on incorrect entity matching.** Despite the high accuracy of legal LLMs in predicting judgment results, we found that a considerable ratio of the confidence score $v(\mathbf{x})$ was mistakenly attributed to interactions on criminal actions made by incorrect entities. In other words, the LLM mistakenly used the criminal action of a person (entity) to make judgment on another unrelated person (entity). To evaluate the impact of such incorrect entity matching on both the SaulLM-7B-Instruct and BAI-Law-13B models, we annotated tokens for criminal actions of unrelated entities as the *forbidden tokens* in $\mathcal{F}$. These forbidden tokens should not influence the judgment for the unrelated entity.

Figure 4 illustrates the test of the SaulLM-7B-Instruct model on the legal case, which showed *Andy bit Charlie, committing an assault, and then Bob hit Charlie with a shovel, leading to murder*. Because tokens such as *"hit,"* *"with a shovel,"* *"injuring,"* and *"death"* described Bob's actions and consequences without a direct relationship with Andy. Thus, these tokens were annotated as forbidden tokens in $\mathcal{F}$. However, we observed that although the SaulLM-7B-Instruct model had used $s^{\text{reliable}} = 21.5\%$ reliable interactions between relevant tokens, such as *"bit"* and *"slightly injured,"* it also modeled a significant number of unreliable interactions containing forbidden tokens *"death"* and *"with a shovel."* However, if we removed these two forbidden tokens for criminal actions of Bob, then the confidence of the judgment of Andy would be significantly affected. This was an obvious representation flaw of the SaulLM-7B-Instruct model.

In comparison, given the same legal case, the BAI-LAW-13B model encoded a ratio of $s^{\text{reliable}} = 22.6\%$ reliable interaction effects, which was a bit higher than a ratio of $s^{\text{reliable}} = 21.5\%$ reliable interactions effects encoded by the SaulLM-7B-Instruct model. In this case, both models primarily relied on unreliable interactions, including forbidden tokens that related to Bob's criminal actions, to make judgment on Andy. For example, the SaulLM-7B-Instruct model used the AND interaction *w.r.t.* the unrelated action $S = \{\text{"with a shovel"}\}$ to contribute 0.93, and the BAI-Law-13B model used the AND interaction $S = \{\text{"death"}\}$ to contribute $-0.43$. This suggested that both legal LLMs handled judgment-related tokens in a local manner, without accurately matching criminal actions with entities. Additional examples of making judgments based on incorrect entity matching are provided in Appendix G.2.

**Problem 3: discrimination in occupation may affect judgments.** We found that the legal LLM usually used interactions on the occupation information to compute the confidence score $v(\mathbf{x})$. This would lead to a significant occupation bias. More interestingly, we discovered that when we replaced the current occupation with another occupation, the interaction containing the occupation token would be significant changed. This indicates a common bias problem, because similar bias may also happen on other attributes (*e.g.*, age, gender, education level, and marital status).

Figure 5 shows the test of the SaulLM-7B-Instruct model on the legal case, in which *Andy, the victim with varying occupations, was robbed of his belongings by two suspicious men*. First, we found that the SaulLM-7B-Instruct model encoded interactions with the occupation tokens "*a judge*," which boosted the confidence of the judgment "*robbery*." More interestingly, if we substituted the occupation tokens "*a judge*" to "*a volunteer*," the interaction between the occupation "*a volunteer*," "*a day's work*," and "*belongings*" decreased from 0.22 to 0.06. This was an important factor that changed the judgment from "*robbery*" to "*not mentioned*." However, if we replaced "*a judge*" with law-related occupations, such as "*a lawyer*" and "*a policeman*," the judgment remained "*robbery*." Besides, the occupation "*a programmer*" changed the judgment to "*not mentioned*." Please see Appendix G.3 for numerical effects of all these occupations. This suggested that the legal LLM sometimes had considerable occupation bias. In comparison, we evaluated the same legal case on the BAI-Law-13B model, as shown in Appendix G.3. Compared to the SaulLM-7B-Instruct model that encoded a ratio $s^{\text{reliable}} = 81.4\%\text{-}84.0\%$ of reliable interaction effects *w.r.t.* different occupations, the BAI-Law-13B model encoded a ratio $s^{\text{reliable}} = 78.9\%\text{-}87.1\%$ of reliable interaction effects. This indicated that both legal LLMs tended to use specific occupational tokens for judgment, instead of correctly analyzing the decision-making logic behind legal judgements. Additional examples of judgments biased by the occupation are provided in Appendix G.3.

**Representation quality of legal LLMs.** Figures 3 and 4 compare the interaction effects of different orders extracted from the SaulLM-7B-Instruct model and the BAI-Law-13B model. We observed that, in both the legal case influenced by unreliable sentimental tokens, and the legal case affected by incorrect entity matching, the BAI-Law-13B model encoded higher order interactions than the SaulLM-7B-Instruct model. This indicated that feature representations of the BAI-Law-13B model was more complex and less generalizable than than those of the SaulLM-7B-Instruct model. In addition, in the legal case that judgments affected by incorrect entity matching, the BAI-Law-13B model encoded a significant number of interactions with negative effects. This suggested that many interactions encoded by the BAI-Law-13B model showed conflicting effects, which was also a sign of over-fitting of the LLM. Tables 1 and 2 in the appendix further show the average ratios of the reliable interaction effects $s_o^{\text{reliable},+}$ and $s_o^{\text{reliable},-}$ for each order $o$ on both LLMs. Experimental results show that while the BAI-Law-13B model encoded more low-order reliable interaction effects, it also encoded more high-order unreliable interaction effects than the SaulLM-7B-Instruct model.

## 4 CONCLUSION

In this paper, we have proposed a method to evaluate the correctness of the detailed decision-making logic of an LLM. The sparsity property and the universal matching property of interactions provide direct mathematical supports for the faithfulness of the interaction-based explanation. Thus, in this paper, we have designed two new metrics to quantify reliable and unreliable interaction effects, according to their alignment with human cognition. Experiments showed that the legal LLMs often relied on a considerable number of problematic interactions to make judgments, even when the judgement prediction was correct. The evaluation of the alignment between the decision-making logic of LLMs and human cognition also contributes to other real applications. For example, it may assist in debugging the hallucination problems, and identifying potential bias behind the language generation results of LLMs.

**Limitations.** Our analysis does not assess the correctness of the numerical scores for interactions, as these scores are often determined by many factors. Positive interactions typically indicate logics that contribute positively to the judgments, while negative interactions may also be intended for other possible correct judgments. Besides, the evaluation based on relevant, irrelavant, and forbidden tokens is only one of the conditions for reliable interactions, and reliable interactions may not always be correct. Nevertheless, this paper presents a precedent for evaluating the correctness of decision-making logic of LLMs.

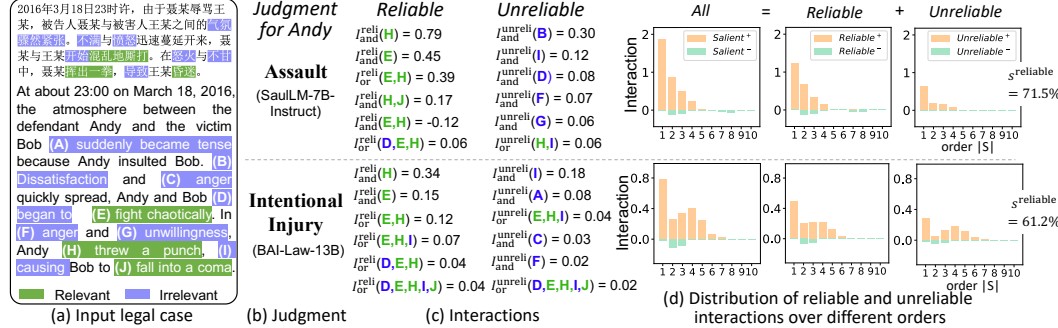

Figure 3: Visualization of judgments influenced by unreliable sentimental tokens. (a) A number of irrelevant tokens were annotated in the legal case, including unreliable sentimental tokens. Criminal actions were annotated as relevant tokens. We also translated the legal case to English as the input of the SaulLM-7B-Instruct model. (b) Judgements predicted by the two legal LLMs, which were both correct according to laws of the two countries. (c,d) We quantified the reliable and unreliable interaction effects of different orders.

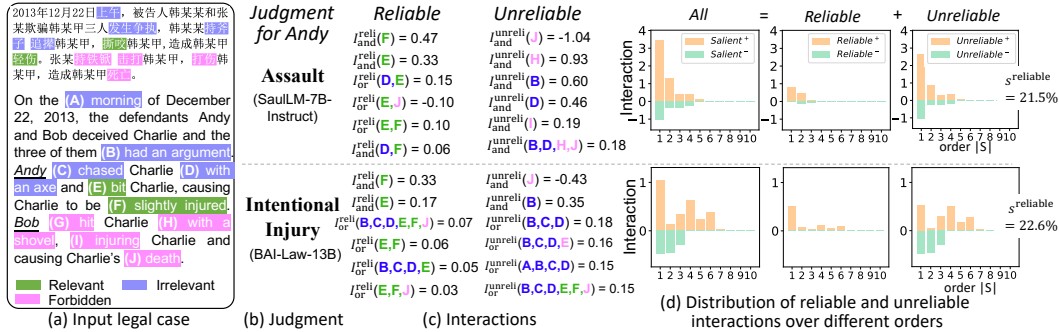

Figure 4: Visualization of judgments affected by incorrect entity matching. (a) A number of irrelevant tokens were annotated in the legal case, including the time and actions that were not the direct reason for the judgment. Criminal actions of the defendant were annotated as relevant tokens. Criminal actions of the unrelated person were annotated as forbidden tokens. (b) Judgements predicted by the two legal LLMs, which were both correct according to laws of the two countries. (c,d) We measured the reliable and unreliable interaction effects of different orders.

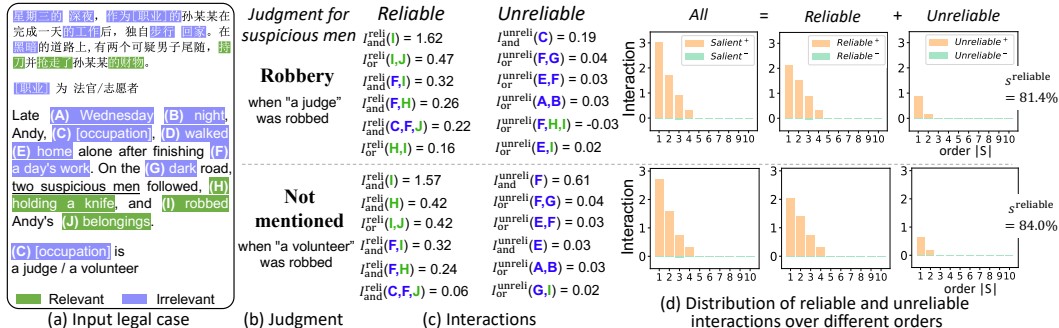

Figure 5: Visualization of judgments biased by discrimination in occupation. (a) A number of irrelevant tokens were annotated in the legal case, including the occupation, time and actions that are not the direct reason for the judgment. Criminal actions of the defendant were annotated as relevant tokens. (b) The SaulLM-7B-Instruct model predicted the judgment based on the legal case with different occupations, respectively. (c,d) We measured the reliable and unreliable interaction effects of different orders. When the occupation was set to "*a judge*," the LLM used 81% reliable interaction effects. In comparison, when the occupation was set to "*a volunteer*," the LLM encoded 84% reliable interaction effects.

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

# A  RELATED WORK

**Factuality and hallucination problems.** Factuality in LLMs refers to whether the language generalization results of LLMs align with the verifiable facts. This includes the ability of LLMs to avoid producing misleading or incorrect information (*i.e.*, factual hallucination), and to effectively generate factually accurate results. For instance, several studies have evaluated the correctness of LLM-generated answers to specific questions (Lin et al., 2021; OpenAI, 2023; Wang et al., 2024). Other works have standardized fact consistency tasks into binary labels, evaluating whether there were factual conflicts within the input text (Honovich et al., 2022). Min et al. (2023) further decomposed language generation results into "atomic" facts, and calculated the proportion of these facts that aligned with a given knowledge source. Additionally, Manakul et al. (2023) introduced a sampling-based method to verify whether LLMs generated factually consistent results, based on the assumption that if an LLM had knowledge of a concept, then the sampled generation results contained consistent factual information.

Hallucination in LLMs typically refers to generated content that is nonsensical or unfaithful to the provided source input (Filippova, 2020; Maynez et al., 2020; Huang et al., 2023). Hallucinations are generally categorized into two primary types, namely intrinsic and extrinsic hallucinations (Maynez et al., 2020; Huang et al., 2021; Dziri et al., 2021; Ji et al., 2023b). Intrinsic hallucinations occur when the generated results contradict the source content, while extrinsic hallucinations arise when the generated results cannot be verified from the provided source. For instance, Bang et al. (2023) found extrinsic hallucinations in ChatGPT's responses, including both untruthful and factual hallucinations, whereas intrinsic hallucinations were rarely observed. OpenAI's latest model, GPT-4 (OpenAI, 2023), has further reduced the model's tendency to hallucinate compared to prior models such as ChatGPT.

**Value alignment.** Value alignment in LLMs aims to ensure LLMs behave in accordance with human intentions and values (Leike et al., 2018; Wang et al., 2023; Ji et al., 2023a). Recent research has focused on improving the ability of LLMs to comprehend instructions, thereby aligning their behavior with human expectations. For instance, OpenAI proposed Supervised Fine-Tuning (SFT) for LLMs, which involved using human-annotated instruction data. LLMs such as InstructGPT (Ouyang et al., 2022) and ChatGPT, both of which employed this technique, have demonstrated significant improvements in understanding human instructions. Ouyang et al. (2022); OpenAI (2023); Touvron et al. (2023) have incorporated the Reinforcement Learning from Human Feedback (RLHF) method to further fine-tune LLMs, enhancing their alignment with human preferences (OpenAI, 2023).

**Using interactions to faithfully explain DNNs.** Ren et al. (2023a) first proposed to quantify interactions between input variables encoded by the DNN, to explain the knowledge in the DNN. Li & Zhang (2023) discovered the discriminative power of interactions between input variables. Ren et al. (2024) further proved that DNNs usually only encoded a small number of interactions. Futhermore, Deng et al. (2024a) found that different attribution scores estimated by fourteen attribution methods, including the Grad-CAM (Selvaraju et al., 2017), Integrated Gradients (Sundararajan et al., 2017), and Shapley value (Lundberg & Lee, 2017)) methods, could all be represented as a combination of interactions. Besides, Zhang et al. (2022) used interactions to explain the mechanism of different methods of boosting adversarial transferability. Ren et al. (2023b) used interactions to define the optimal baseline value for computing Shapley values. Deng et al. (2022) found that for most DNNs it was difficult to learn interactions with median number of input variables, and it was discovered that DNNs and Bayesian neural networks were unlikely to model complex interactions with many input variables (Ren et al., 2023c; Liu et al., 2024). Zhou et al. (2024) used the generalization power of different interactions to explain the generalization power of DNNs.

Unlike evaluations on language generation results, we propose a method that leverages interaction-based explanations to evaluate the correctness of decision-making logic encoded by a LLM. This approach enables us to evaluate the alignment between the decision-making logic of LLMs and human cognition.

# B  PROOF OF THEOREM

**Theorem 1** (Universal matching property)  Given an input sample $\mathbf{x}$, the network output score $v(\mathbf{x}_T) \in \mathbb{R}$ on each masked sample $\{\mathbf{x}_T | T \subseteq N\}$ can be well matched by a surrogate logical model

$h(\mathbf{x}_T)$ on each masked sample $\{\mathbf{x}_T | T \subseteq N\}$. The surrogate logical model $h(\mathbf{x}_T)$ uses the sum of AND interactions and OR interactions to accurately fit the network output score $v(\mathbf{x}_T)$.

$$\forall T \subseteq N, v(\mathbf{x}_T) = h(\mathbf{x}_T).$$

$$h(\mathbf{x}_T) = v(\mathbf{x}_\emptyset) + \sum_{S \subseteq N, S \neq \emptyset} \mathbb{1}(\begin{smallmatrix} \mathbf{x}_T \text{ triggers} \\ \text{AND relation } S \end{smallmatrix}) \cdot I_{\text{and}}(S|\mathbf{x}_T) + \mathbb{1}(\begin{smallmatrix} \mathbf{x}_T \text{ triggers} \\ \text{OR relation } S \end{smallmatrix}) \cdot I_{\text{or}}(S|\mathbf{x}_T)$$

$$= v(\mathbf{x}_\emptyset) + \underbrace{\sum_{S \subseteq T, S \neq \emptyset} I_{\text{and}}(S|\mathbf{x}_T)}_{v_{\text{and}}(\mathbf{x}_T)} + \underbrace{\sum_{S \subseteq N, S \cap T \neq \emptyset} I_{\text{or}}(S|\mathbf{x}_T)}_{v_{\text{or}}(\mathbf{x}_T)} \quad (8)$$

Let us set a surrogate logical model $h(\mathbf{x}_T) = v(\mathbf{x}_T), \forall T \subseteq N$, which utilizes the sum of AND interactions $I_{\text{and}}(S|\mathbf{x})$ and OR interactions $I_{\text{or}}(S|\mathbf{x})$ in Equation (2) to fit the network output score $v(\mathbf{x}_T)$, *i.e.*, $v(\mathbf{x}_T) = h(\mathbf{x}_T) = v_{\text{and}}(\mathbf{x}_T) + v_{\text{or}}(\mathbf{x}_T)$.

To be specific, we use the sum of AND interactions $I_{\text{and}}(S|\mathbf{x})$ to compute the component for AND interactions $v_{\text{and}}(\mathbf{x}_T)$, *i.e.*, $v_{\text{and}}(\mathbf{x}_T) = \sum_{S \subseteq T} I_{\text{and}}(S|\mathbf{x}_T)$. Then, we use the sum of OR interactions $I_{\text{or}}(S|\mathbf{x})$ to compute the component for OR interactions $v_{\text{or}}(\mathbf{x}_T)$, *i.e.*, $v_{\text{or}}(\mathbf{x}_T) = \sum_{S \subseteq N, S \cap T \neq \emptyset} I_{\text{or}}(S|\mathbf{x}_T)$. Finally, we use the sum of AND-OR interactions to fit the network output score, *i.e.*, $v(\mathbf{x}_T) = h(\mathbf{x}_T) = v_{\text{and}}(\mathbf{x}_T) + v_{\text{or}}(\mathbf{x}_T)$.

**(1) Universal matching property of AND interactions.**

Ren et al. (2023a) have used the Haranyi dividend (Harsanyi, 1963) $I_{\text{and}}(S|\mathbf{x})$ to state the universal matching property of AND interactions. The output of a well-trained DNN on all $2^n$ masked samples $\{\mathbf{x}_T | T \subseteq N\}$ could be universally explained by the all interaction primitives in $T \subseteq N$, *i.e.*, $\forall T \subseteq N, v_{\text{and}}(\mathbf{x}_T) = \sum_{S \subseteq T} I_{\text{and}}(S|\mathbf{x})$.

Specifically, the AND interaction (as known as Harsanyi dividend) is defined as $I_{\text{and}}(S|\mathbf{x}) := \sum_{L \subseteq S}(-1)^{|S|-|L|} v_{\text{and}}(\mathbf{x}_L)$ in Equation (2). To compute the sum of AND interactions $\forall T \subseteq N, \sum_{S \subseteq T} I_{\text{and}}(S|\mathbf{x}) = \sum_{S \subseteq T} \sum_{L \subseteq S}(-1)^{|S|-|L|} v_{\text{and}}(\mathbf{x}_L)$, we first exchange the order of summation of the set $L \subseteq S \subseteq T$ and the set $S \supseteq L$. That is, we compute all linear combinations of all sets $S$ containing $L$ with respect to the model outputs $v_{\text{and}}(\mathbf{x}_L)$, given a set of input variables $L$, *i.e.*, $\sum_{S:L \subseteq S \subseteq T}(-1)^{|S|-|L|} v_{\text{and}}(\mathbf{x}_L)$. Then, we compute all summations over the set $L \subseteq T$.

In this way, we can compute them separately for different cases of $L \subseteq S \subseteq T$. In the following, we consider the cases (1) $L = S = T$, and (2) $L \subseteq S \subseteq T, L \neq T$, respectively.

(1) When $L = S = T$, the linear combination of all subsets $S$ containing $L$ with respect to the model output $v_{\text{and}}(\mathbf{x}_L)$ is $(-1)^{|T|-|T|} v_{\text{and}}(\mathbf{x}_L) = v_{\text{and}}(\mathbf{x}_L)$.

(2) When $L \subseteq S \subseteq T, L \neq T$, the linear combination of all subsets $S$ containing $L$ with respect to the model output $v_{\text{and}}(\mathbf{x}_L)$ is $\sum_{S:L \subseteq S \subseteq T}(-1)^{|S|-|L|} v_{\text{and}}(\mathbf{x}_L)$. For all sets $S : T \supseteq S \supseteq L$, let us consider the linear combinations of all sets $S$ with number $|S|$ for the model output $v_{\text{and}}(\mathbf{x}_L)$, respectively. Let $m := |S| - |L|, (0 \leq m \leq |T| - |L|)$, then there are a total of $C_{|T|-|L|}^m$ combinations of all sets $S$ of order $|S|$. Thus, given $L$, accumulating the model outputs $v_{\text{and}}(\mathbf{x}_L)$ corresponding to all $S \supseteq L$, then $\sum_{S:L \subseteq S \subseteq T}(-1)^{|S|-|L|} v_{\text{and}}(\mathbf{x}_L) = v_{\text{and}}(\mathbf{x}_L) \cdot \underbrace{\sum_{m=0}^{|T|-|L|} C_{|T|-|L|}^m (-1)^m}_{=0} = 0$.

Please see the complete derivation of the following formula.

$$\sum_{S \subseteq T} I_{\text{and}}(S|\mathbf{x}_T) = \sum_{S \subseteq T} \sum_{L \subseteq S}(-1)^{|S|-|L|} v_{\text{and}}(\mathbf{x}_L)$$

$$= \sum_{L \subseteq T} \sum_{S:L \subseteq S \subseteq T}(-1)^{|S|-|L|} v_{\text{and}}(\mathbf{x}_L)$$

$$= \underbrace{v_{\text{and}}(\mathbf{x}_T)}_{L=T} + \sum_{L \subseteq T, L \neq T} v_{\text{and}}(\mathbf{x}_L) \cdot \underbrace{\sum_{m=0}^{|T|-|L|} C_{|T|-|L|}^m (-1)^m}_{=0} \quad (9)$$

$$= v_{\text{and}}(\mathbf{x}_T).$$

Furthermore, we can understand the above equation in a physical sense. Given a masked sample $\mathbf{x}_T$, if $\mathbf{x}_T$ triggers an AND relationship $S$ (the co-appearance of all input variables in $S$), then $S \subseteq T$. Thus, we accumulate the interaction effects $I_{\text{and}}(S|\mathbf{x})$ of any AND relationship $S$ triggered by $\mathbf{x}_T$ as follows,

$$
\begin{aligned}
v(\mathbf{x}_\emptyset) + \sum_{S \subseteq N, S \neq \emptyset} \mathbb{1}(\begin{smallmatrix} \mathbf{x}_T \text{ triggers} \\ \text{AND relation } S \end{smallmatrix}) \cdot I_{\text{and}}(S|\mathbf{x}_T) & \\
= v(\mathbf{x}_\emptyset) + \sum_{S \subseteq T, S \neq \emptyset} I_{\text{and}}(S|\mathbf{x}_T) & \\
= \sum_{S \subseteq T} I_{\text{and}}(S|\mathbf{x}_T) \qquad //I_{\text{and}}(\emptyset|\mathbf{x}_T) = v_{\text{and}}(\mathbf{x}_\emptyset) = v(\mathbf{x}_\emptyset) & \\
= v_{\text{and}}(\mathbf{x}_T). &
\end{aligned}
\tag{10}
$$

**(2) Universal matching property of OR interactions.**

According to the definition of OR interactions in Equation (2), we will derive that $\forall T \subseteq N, v_{\text{or}}(\mathbf{x}_T) = \sum_{S \subseteq N, S \cap T \neq \emptyset} I_{\text{or}}(S|\mathbf{x}_T), s.t., I_{\text{or}}(\emptyset|\mathbf{x}_T) = v_{\text{or}}(\mathbf{x}_\emptyset) = 0$.

Specifically, the OR interaction is defined as $I_{\text{or}}(S|\mathbf{x}) := -\sum_{L \subseteq S}(-1)^{|S|-|L|}v_{\text{or}}(\mathbf{x}_{N \setminus L})$ in Equation (2). To compute the sum of OR interactions $\forall T \subseteq N, \sum_{S \subseteq N, S \cap T \neq \emptyset} I_{\text{or}}(S|\mathbf{x}_T) = \sum_{S \subseteq N, S \cap T \neq \emptyset} \left[ -\sum_{L \subseteq S}(-1)^{|S|-|L|}v_{\text{or}}(\mathbf{x}_{N \setminus L}) \right]$, we first exchange the order of summation of the set $L \subseteq S \subseteq N$ and the set $S \cap T \neq \emptyset$. That is, we compute all linear combinations of all sets $S$ containing $L$ with respect to the model outputs $v_{\text{or}}(\mathbf{x}_{N \setminus L})$, given a set of input variables $L$, i.e., $\sum_{S \cap T \neq \emptyset, N \supseteq S \supseteq L}(-1)^{|S|-|L|}v_{\text{or}}(\mathbf{x}_{N \setminus L})$. Then, we compute all summations over the set $L \subseteq N$.

In this way, we can compute them separately for different cases of $L \subseteq S \subseteq N, S \cap T \neq \emptyset$. In the following, we consider the cases (1) $L = N \setminus T$, (2) $L = N$, (3) $L \cap T \neq \emptyset, L \neq N$, and (4) $L \cap T = \emptyset, L \neq N \setminus T$, respectively.

(1) When $L = N \setminus T$, the linear combination of all subsets $S$ containing $L$ with respect to the model output $v_{\text{or}}(\mathbf{x}_{N \setminus L})$ is $\sum_{S \cap T \neq \emptyset, S \supseteq L}(-1)^{|S|-|L|}v_{\text{or}}(\mathbf{x}_{N \setminus L}) = \sum_{S \cap T \neq \emptyset, S \supseteq L}(-1)^{|S|-|L|}v_{\text{or}}(\mathbf{x}_T)$. For all sets $S \supseteq L, S \cap T \neq \emptyset$ (then $S \neq N \setminus T, S \neq L$), let us consider the linear combinations of all sets $S$ with number $|S|$ for the model output $v_{\text{or}}(\mathbf{x}_T)$, respectively. Let $|S'| := |S|-|L|, (1 \leq |S'| \leq |T|)$, then there are a total of $C_{|T|}^{|S'|}$ combinations of all sets $S$ of order $|S|$. Thus, given $L$, accumulating the model outputs $v_{\text{or}}(\mathbf{x}_T)$ corresponding to all $S \supseteq L$, then $\sum_{S \cap T \neq \emptyset, S \supseteq L}(-1)^{|S|-|L|}v_{\text{or}}(\mathbf{x}_{N \setminus L}) = v_{\text{or}}(\mathbf{x}_T) \cdot \underbrace{\sum_{|S'|=1}^{|T|} C_{|T|}^{|S'|}(-1)^{|S'|}}_{=-1} = -v_{\text{or}}(\mathbf{x}_T)$.

(2) When $L = N$ (then $S = N$), the linear combination of all subsets $S$ containing $L$ with respect to the model output $v_{\text{or}}(\mathbf{x}_{N \setminus L})$ is $\sum_{S \cap T \neq \emptyset, S \supseteq L}(-1)^{|S|-|L|}v_{\text{or}}(\mathbf{x}_{N \setminus L}) = (-1)^{|N|-|N|}v_{\text{or}}(\mathbf{x}_\emptyset) = v_{\text{or}}(\mathbf{x}_\emptyset) = 0, (I_{\text{or}}(\emptyset|\mathbf{x}_T) = v_{\text{or}}(\mathbf{x}_\emptyset) = 0)$.

(3) When $L \cap T \neq \emptyset, L \neq N$, the linear combination of all subsets $S$ containing $L$ with respect to the model output $v_{\text{or}}(\mathbf{x}_{N \setminus L})$ is $\sum_{S \cap T \neq \emptyset, S \supseteq L}(-1)^{|S|-|L|}v_{\text{or}}(\mathbf{x}_{N \setminus L})$. For all sets $S \supseteq L, S \cap T \neq \emptyset$, let us consider the linear combinations of all sets $S$ with number $|S|$ for the model output $v_{\text{or}}(\mathbf{x}_T)$, respectively. Let us split $|S| - |L|$ into $|S'|$ and $|S''|$, i.e., $|S| - |L| = |S'| + |S''|$, where $S' = \{i|i \in S, i \notin L, i \in N \setminus T\}$, $S'' = \{i|i \in S, i \notin L, i \in T\}$ (then $0 \leq |S''| \leq |T| - |T \cap L|$) and $S' + S'' + L = S$. In this way, there are a total of $C_{|T|-|T \cap L|}^{|S''|}$ combinations of all sets $S''$ of order $|S''|$. Thus, given $L$, accumulating the model outputs $v_{\text{or}}(\mathbf{x}_{N \setminus L})$ corresponding to all $S \supseteq L$, then $\sum_{S \cap T \neq \emptyset, S \supseteq L}(-1)^{|S|-|L|}v_{\text{or}}(\mathbf{x}_{N \setminus L}) = v_{\text{or}}(\mathbf{x}_{N \setminus L}) \cdot \sum_{S' \subseteq N \setminus T \setminus L} \underbrace{\sum_{|S''|=0}^{|T|-|T \cap L|} C_{|T|-|T \cap L|}^{|S''|}(-1)^{|S'|+|S''|}}_{=0} = 0$.

(4) When $L \cap T = \emptyset, L \neq N \setminus T$, the linear combination of all subsets $S$ containing $L$ with respect to the model output $v_{\text{or}}(\mathbf{x}_{N \setminus L})$ is $\sum_{S:S \cap T \neq \emptyset, S \supseteq L}(-1)^{|S|-|L|}v_{\text{or}}(\mathbf{x}_{N \setminus L})$. Similarly, let us

split $|S|-|L|$ into $|S'|$ and $|S''|$, $i.e.,|S|-|L| = |S'|+|S''|$, where $S' = \{i|i \in S, i \notin L, i \in N\setminus T\}$, $S'' = \{i|i \in S, i \in T\}$ (then $0 \leq |S''| \leq |T|$) and $S' + S'' + L = S$. In this way, there are a total of $C_{|T|}^{|S''|}$ combinations of all sets $S''$ of order $|S''|$. Thus, given $L$, accumulating the model outputs $v_{\mathrm{or}}(\mathbf{x}_{N\setminus L})$ corresponding to all $S \supseteq L$, then $\sum_{S\cap T\neq\emptyset, S\supseteq L}(-1)^{|S|-|L|}v_{\mathrm{or}}(\mathbf{x}_{N\setminus L}) = $
$v_{\mathrm{or}}(\mathbf{x}_{N\setminus L}) \cdot \sum_{S'\subseteq N\setminus T\setminus L} \underbrace{\sum_{|S''|=0}^{|T|} C_{|T|}^{|S''|}(-1)^{|S'|+|S''|}}_{=0} = 0.$

Please see the complete derivation of the following formula.

$$\sum_{S\subseteq N, S\cap T\neq\emptyset} I_{\mathrm{or}}(S|\mathbf{x}_T) = \sum_{S\subseteq N, S\cap T\neq\emptyset}\left[-\sum_{L\subseteq S}(-1)^{|S|-|L|}v_{\mathrm{or}}(\mathbf{x}_{N\setminus L})\right]$$

$$= -\sum_{L\subseteq N}\sum_{S\cap T\neq\emptyset, N\supseteq S\supseteq L}(-1)^{|S|-|L|}v_{\mathrm{or}}(\mathbf{x}_{N\setminus L})$$

$$= -\left[\sum_{|S'|=1}^{|T|}C_{|T|}^{|S'|}(-1)^{|S'|}\right]\cdot\underbrace{v_{\mathrm{or}}(\mathbf{x}_T)}_{L=N\setminus T} - \underbrace{v_{\mathrm{or}}(\mathbf{x}_{\emptyset})}_{L=N}$$

$$-\sum_{L\cap T\neq\emptyset, L\neq N}\left[\sum_{S'\subseteq N\setminus T\setminus L}\left(\sum_{|S''|=0}^{|T|-|T\cap L|}C_{|T|-|T\cap L|}^{|S''|}(-1)^{|S'|+|S''|}\right)\right]\cdot v_{\mathrm{or}}(\mathbf{x}_{N\setminus L})$$

$$-\sum_{L\cap T=\emptyset, L\neq N\setminus T}\left[\sum_{S'\subseteq N\setminus T\setminus L}\left(\sum_{|S''|=0}^{|T|}C_{|T|}^{|S''|}(-1)^{|S'|+|S''|}\right)\right]\cdot v_{\mathrm{or}}(\mathbf{x}_{N\setminus L})$$

$$= -(-1)\cdot v_{\mathrm{or}}(\mathbf{x}_T) - v_{\mathrm{or}}(\mathbf{x}_{\emptyset}) - \sum_{L\cap T\neq\emptyset, L\neq N}\left[\sum_{S'\subseteq N\setminus T\setminus L}0\right]\cdot v_{\mathrm{or}}(\mathbf{x}_{N\setminus L})$$

$$-\sum_{L\cap T=\emptyset, L\neq N\setminus T}\left[\sum_{S'\subseteq N\setminus T\setminus L}0\right]\cdot v_{\mathrm{or}}(\mathbf{x}_{N\setminus L})$$

$$= v_{\mathrm{or}}(\mathbf{x}_T) \tag{11}$$

Furthermore, we can understand the above equation in a physical sense. Given a masked sample $\mathbf{x}_T$, if $\mathbf{x}_T$ triggers an OR relationship $S$ (the presence of any input variable in $S$), then $S\cap T \neq \emptyset, S \subseteq N$. Thus, we accumulate the interaction effects $I_{\mathrm{or}}(S|\mathbf{x})$ of any OR relationship $S$ triggered by $\mathbf{x}_T$ as follows,

$$\sum_{S\subseteq N, S\neq\emptyset}\mathbb{1}\binom{\mathbf{x}_T \text{ triggers}}{\text{OR relation } S}\cdot I_{\mathrm{or}}(S|\mathbf{x}_T)$$
$$= \sum_{S\subseteq N, S\cap T\neq\emptyset}I_{\mathrm{or}}(S|\mathbf{x}_T) \tag{12}$$
$$= v_{\mathrm{or}}(\mathbf{x}_T).$$

**(3) Universal matching property of AND-OR interactions.**

With the universal matching property of AND interactions and the universal matching property of OR interactions, we can easily get $v(\mathbf{x}_T) = h(\mathbf{x}_T) = v_{\mathrm{and}}(\mathbf{x}_T) + v_{\mathrm{or}}(\mathbf{x}_T) = v(\mathbf{x}_{\emptyset}) + \sum_{S\subseteq T, S\neq\emptyset}I_{\mathrm{and}}(S|\mathbf{x}_T) + \sum_{S\subseteq N, S\cap T\neq\emptyset}I_{\mathrm{or}}(S|\mathbf{x}_T)$, thus, we obtain the universal matching property of AND-OR interactions.

## C  PROOF OF LEMMA

**Lemma 1** (Sparsity property) The surrogate logical model $h(\mathbf{x}_T)$ on each randomly masked sample $\mathbf{x}_T, T \subseteq N$ mainly uses the sum of a small number of salient AND interactions and salient OR interactions to approximate the network output score $v(\mathbf{x}_T)$.

$$v(\mathbf{x}_T) = h(\mathbf{x}_T) \approx v(\mathbf{x}_{\emptyset}) + \sum_{S\in\Omega^{\mathrm{and}}}\mathbb{1}\binom{\mathbf{x}_T \text{ triggers}}{\text{AND relation } S}\cdot I_{\mathrm{and}}(S|\mathbf{x}_T) + \sum_{S\in\Omega^{\mathrm{or}}}\mathbb{1}\binom{\mathbf{x}_T \text{ triggers}}{\text{OR relation } S}\cdot I_{\mathrm{or}}(S|\mathbf{x}_T)$$
$$\tag{13}$$

Ren et al. (2024) have proven that under some common conditions[6], the confidence score $v_{\text{and}}(\mathbf{x}_T)$ of a well-trained DNN on all $2^n$ masked samples $\{\mathbf{x}_T | T \subseteq N\}$ could be universally approximated by a small number of AND interactions $T \in \Omega^{\text{and}}$ with salient interaction effects $I_{\text{and}}(T|\mathbf{x})$, $s.t.$, $|\Omega^{\text{and}}| \ll 2^n$, $i.e.$, $\forall T \subseteq N, v_{\text{and}}(\mathbf{x}_T) = \sum_{S \subseteq T} I_{\text{and}}(S|\mathbf{x}) \approx \sum_{S \subseteq T: S \in \Omega^{\text{and}}} I_{\text{and}}(S|\mathbf{x})$.

According to Equation (10), $v_{\text{and}}(\mathbf{x}_T) = \sum_{S \subseteq T} I_{\text{and}}(S|\mathbf{x}) = v(\mathbf{x}_\emptyset) + \sum_{S \subseteq N, S \neq \emptyset} \mathbb{1}(\substack{\mathbf{x}_T \text{ triggers} \\ \text{AND relation } S}) \cdot I_{\text{and}}(S|\mathbf{x}_T)$. Therefore, $v_{\text{and}}(\mathbf{x}_T) \approx v(\mathbf{x}_\emptyset) + \sum_{S \in \Omega^{\text{and}}} \mathbb{1}(\substack{\mathbf{x}_T \text{ triggers} \\ \text{AND relation } S}) \cdot I_{\text{and}}(S|\mathbf{x}_T)$.

Besides, as proven in Appendix D, the OR interaction can be considered as a specific AND interaction. Thus, the confidence score $v_{\text{or}}(\mathbf{x}_T)$ of a well-trained DNN on all $2^n$ masked samples $\{\mathbf{x}_T | T \subseteq N\}$ could be universally approximated by a small number of OR interactions $T \in \Omega^{\text{or}}$ with salient interaction effects $I_{\text{or}}(T|\mathbf{x})$, $s.t.$, $|\Omega^{\text{or}}| \ll 2^n$. Similarly, $v_{\text{or}}(\mathbf{x}_T) = \sum_{S \subseteq N, S \neq \emptyset} \mathbb{1}(\substack{\mathbf{x}_T \text{ triggers} \\ \text{OR relation } S}) \cdot I_{\text{or}}(S|\mathbf{x}_T) \approx \sum_{S \in \Omega^{\text{or}}} \mathbb{1}(\substack{\mathbf{x}_T \text{ triggers} \\ \text{OR relation } S}) \cdot I_{\text{or}}(S|\mathbf{x}_T)$.

In this way, the surrogate logical model $h(\mathbf{x}_T)$ on each randomly masked sample $\mathbf{x}_T, T \subseteq N$ mainly uses the sum of a small number of salient AND interactions and salient OR interactions to approximate the network output score $v(\mathbf{x}_T)$, $i.e.$, $v(\mathbf{x}_T) = h(\mathbf{x}_T) = v_{\text{and}}(\mathbf{x}_T) + v_{\text{or}}(\mathbf{x}_T) \approx v(\mathbf{x}_\emptyset) + \sum_{S \in \Omega^{\text{and}}} \mathbb{1}(\substack{\mathbf{x}_T \text{ triggers} \\ \text{AND relation } S}) \cdot I_{\text{and}}(S|\mathbf{x}_T) + \sum_{S \in \Omega^{\text{or}}} \mathbb{1}(\substack{\mathbf{x}_T \text{ triggers} \\ \text{OR relation } S}) \cdot I_{\text{or}}(S|\mathbf{x}_T)$.

## D  OR INTERACTIONS CAN BE CONSIDERED SPECIFIC AND INTERACTIONS

The OR interaction $I_{\text{or}}(S|\mathbf{x})$ can be considered as a specific AND interaction $I_{\text{and}}(S|\mathbf{x})$, if we inverse the definition of the masked state and the unmasked state of an input variable.

Given a DNN $v: \mathbb{R}^n \to \mathbb{R}$ and an input sample $\mathbf{x} \in \mathbb{R}^n$, if we arbitrarily mask the input sample, we can get $2^n$ different masked samples $\mathbf{x}_S, \forall S \subseteq N$. Specifically, let us use baseline values $\mathbf{b} \in \mathbb{R}^n$ to represent the masked state of a masked sample $\mathbf{x}_S$, $i.e.$,

$$(\mathbf{x}_S)_i = \begin{cases} x_i, & i \in S \\ b_i, & i \notin S \end{cases} \tag{14}$$

Conversely, if we inverse the definition of the masked state and the unmasked state of an input variable, $i.e.$, we consider $\mathbf{b}$ as the input sample, and consider the original value $\mathbf{x}$ as the masked state, then the masked sample $\mathbf{b}_S$ can be defined as follows.

$$(\mathbf{b}_S)_i = \begin{cases} b_i, & i \in S \\ x_i, & i \notin S \end{cases} \tag{15}$$

According to the above definition of a masked sample in Equations (14) and (15), we can get $\mathbf{x}_{N \setminus S} = \mathbf{b}_S$. To simply the analysis, if we assume that $v_{\text{and}}(\mathbf{x}_T) = v_{\text{or}}(\mathbf{x}_T) = 0.5v(\mathbf{x}_T)$, then the OR interaction $I_{\text{or}}(S|\mathbf{x})$ in Equation (2) can be regarded as a specific AND interaction $I_{\text{and}}(S|\mathbf{b})$ as follows.

$$\begin{aligned} I_{\text{or}}(S|\mathbf{x}) &= -\sum_{T \subseteq S} (-1)^{|S|-|T|} v_{\text{or}}(\mathbf{x}_{N \setminus T}), \\ &= -\sum_{T \subseteq S} (-1)^{|S|-|T|} v_{\text{or}}(\mathbf{b}_T), \\ &= -\sum_{T \subseteq S} (-1)^{|S|-|T|} v_{\text{and}}(\mathbf{b}_T), \\ &= -I_{\text{and}}(S|\mathbf{b}). \end{aligned} \tag{16}$$

## E  GENERALIZATION POWER OF INTERACTIONS OVER DIFFERENT ORDERS

In this section, we will give the definition and quantification of the generalization power of interactions over different orders. The generalization power of an interaction is defined as the transferability

---

[6]There are three assumptions. (1) The high order derivatives of the DNN output with respect to the input variables are all zero. (2) The DNN works well on the masked samples, and yield higher confidence when the input sample is less masked. (3) The confidence of the DNN does not drop significantly on the masked samples.

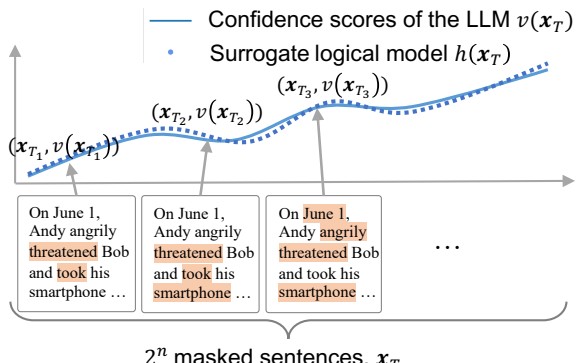 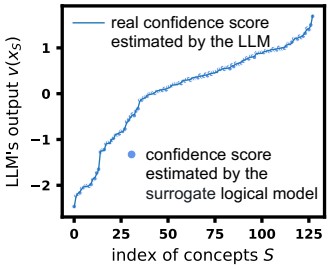

(a) Sketch map of the universal-matching property

(b) Experimental verification: the surrogate logical model $h(\boldsymbol{x}_T)$ well fits the confidence scores of the LLM $v(\boldsymbol{x}_T)$

Figure 6: (a) Illustration of universal-matching property of the extracted interactions. (b) Experiment verifies that the surrogate logical model $h(\mathbf{x}_T)$ can accurately fit the confidence scores of the LLM $v(\mathbf{x}_T)$ on all $2^n$ masked samples $\{\mathbf{x}_T | T \subseteq N\}$, i.e., $\forall T \subseteq N, v(\mathbf{x}_T) = h(\mathbf{x}_T)$, no matter how we randomly mask the input sample $\mathbf{x}$ in $2^n$ different masking states $T \subseteq N$.

---

**Algorithm 1** Computing AND-OR interactions

1: **Input:** Input legal case $\mathbf{x}$, the legal LLM $v(\cdot)$, and the annotations of the relevant, irrelevant, and forbidden tokens in $\mathbf{x}$.
2: **Output:** A set of reliable interactions $I_{\text{and}}^{\text{reliable}}(S|\mathbf{x})$ and $I_{\text{or}}^{\text{reliable}}(S|\mathbf{x})$, and the ratio of reliable interaction effects $s^{\text{reliable}}$
3: Input the legal case $\mathbf{x}$ into the legal LLM, and generate the judgment (a sequence of tokens);
4: **for** $S \subseteq N$ **do**
5:     For each masked sample $\mathbf{x}_S$, compute the confidence score $v(\mathbf{x}_S)$ based on Equation (1);
6: **end for**
7: **for** $S \subseteq N$ **do**
8:     Given $v(\mathbf{x}_S)$ for all combinations $S \subseteq N$, compute each AND interaction $I_{\text{and}}(S|\mathbf{x})$ and each OR interaction $I_{\text{or}}(S|\mathbf{x})$ via $\min_{\{\gamma_T\}} \sum_{S \subseteq N, S \neq \emptyset}[|I_{\text{and}}(S|\mathbf{x})| + |I_{\text{or}}(S|\mathbf{x})|]$;
9: **end for**
10: **for** $S \subseteq N$ **do**
11:     Compute the reliable AND interaction effect $I_{\text{and}}^{\text{reliable}}(S|\mathbf{x})$ and the reliable OR interaction effect $I_{\text{or}}^{\text{reliable}}(S|\mathbf{x})$ based on Equations (5) and (6).
12: **end for**
13: Compute the ratio of reliable interaction effects $s^{\text{reliable}}$ based on Equation (7);
14: **return** $I_{\text{and}}^{\text{reliable}}(S|\mathbf{x})$, $I_{\text{or}}^{\text{reliable}}(S|\mathbf{x})$, $s^{\text{reliable}}$

---

of this interaction from training samples to test samples. Specifically, if an interaction pattern $S \subseteq N$ frequently occurs in the training set, but rarely appears in the test set, then the interaction pattern $S$ exhibits low generalization power. Conversely, if an interaction pattern $S$ consistently appears in both the training and test sets, it demonstrates high generalization power.

Specifically, for a given classification task, Zhou et al. (2024) defined the generalization power of $m$-order interactions *w.r.t.* the category $c$ as the Jaccard similarity between the interactions observed in the training samples and those in the test samples for each category $c$.

$$\text{sim}(\hat{I}_{\text{train},c}^{(m)}, \hat{I}_{\text{test},c}^{(m)}) = \frac{\| \min(\hat{I}_{\text{train},c}^{(m)}, \hat{I}_{\text{test},c}^{(m)}) \|_1}{\| \max(\hat{I}_{\text{train},c}^{(m)}, \hat{I}_{\text{test},c}^{(m)}) \|_1} \tag{17}$$

where $\hat{I}_{\text{train},c}^{(m)} = [(\max(I_{\text{train},c}^{(m)}, 0))^{\intercal}, (-\min(I_{\text{train},c}^{(m)}, 0))^{\intercal}]^{\intercal} \in \mathbb{R}^{2d}$ is conducted from $I_{\text{train},c}^{(m)}$ to ensure that all elements are non-negative. Here, $I_{\text{train},c}^{(m)} = [I_{\text{train},c}^{(m)}(S_1), I_{\text{train},c}^{(m)}(S_2), \cdots, I_{\text{train},c}^{(m)}(S_d)]^{\intercal} \in \mathbb{R}^d$ represents the distribution of $m$-order interactions over the training samples for category $c$, where $d = C_n^m$ enumerates all possible $m$-order interactions. Specifically, $I_{\text{train},c}^{(m)}(S_i) =$

Table 1: Average ratio (%) of reliable interaction effects of each order on the SaulLM-7B-Instruct model.

| order | 1 | 2 | 3 | 4 | 5 | 6 | 7 | 8 | 9 | 10 |
|---|---|---|---|---|---|---|---|---|---|---|
| $s_o^{\text{reliable},+}$ | 60.19 | 56.53 | 49.51 | 48.86 | 43.74 | 30.92 | 42.86 | NAN | NAN | NAN |
| $s_o^{\text{reliable},-}$ | NAN | 66.79 | 53.89 | 58.67 | 50.34 | 35.20 | 52.38 | NAN | NAN | NAN |

Table 2: Average ratio (%) of reliable interaction effects of each order on the BAI-Law-13B model.

| order | 1 | 2 | 3 | 4 | 5 | 6 | 7 | 8 | 9 | 10 |
|---|---|---|---|---|---|---|---|---|---|---|
| $s_o^{\text{reliable},+}$ | 56.22 | 71.24 | 49.02 | 49.17 | 46.56 | 40.10 | 31.70 | 25.00 | 22.22 | NAN |
| $s_o^{\text{reliable},-}$ | 58.15 | 71.06 | 69.68 | 63.02 | 49.73 | 35.98 | 42.86 | NAN | NAN | NAN |

$\mathbb{E}_{\mathbf{x} \in \mathcal{D}_{\text{train},c}}[I(S_i|\mathbf{x})]$ denotes the average interaction effect of the set $S_i$ across different training samples within category $c$.

Therefore, for each category $c$, a high similarity $\text{sim}(\hat{I}_{\text{train},c}^{(m)}, \hat{I}_{\text{test},c}^{(m)})$ indicates that most $m$-order interactions from the training samples generalize well to the test samples.

Using the average similarity over different categories, *i.e.*, similarity $= \mathbb{E}_c[\text{sim}(\hat{I}_{\text{train},c}^{(m)}, \hat{I}_{\text{test},c}^{(m)})]$, Zhou et al. (2024) have empirically found that the low-order interactions usually exhibit stronger generalization power than high-order interactions. Specifically, Figure 4 in (Zhou et al., 2024) shows that compared to high order interaction patterns, DNNs are more likely to extract similar low order interaction patterns from both training and test data.

# F    ACCURACY OF THE LEGAL LLM

Colombo et al. (2024) reported the accuracy of the SaulLM-7B-Instruct model, which achieved state-of-the-art results among 7B models, within the legal domain. Specifically, they followed (Guha et al., 2023) to use *balanced accuracy* as the metric. Balanced accuracy shows its strength for handling imbalanced classification tasks. They tested the balanced accuracy on two popular benchmarks, *i.e.*, the LegalBench-Instruct benchmark (Guha et al., 2023) and the Massive Multitask Language Understanding (MMLU) benchmark (Hendrycks et al., 2021). The LegalBench-Instruct benchmark is a supplemental iteration of LegalBench (Guha et al., 2023), designed to evaluate the legal proficiency of LLMs. To further evaluate the performance of LLMs in legal contexts, the authors incorporated legal tasks from the MMLU benchmark, focusing specifically on the international law, professional law and jurisprudence.

Colombo et al. (2024) compared the SaulLM-7B-Instruct model to other 7B and 13B open-source models, including Mistral-7B (Jiang et al., 2023) and the Llama2 family (Touvron et al., 2023). Table 4 shows that SaulLM-7B-Instruct achieved state-of-the-art performance on the LegalBench-Instruct benchmark, outperforming its competitors in the legal domain.

Table 3: Comparison of LLMs on the LegalBench-Instruct benchmark.

| LLMs | SaulLM-7B-Instruct | Mistral-7B-v1 | Mistral-7B-v2 | Llama2-13B-chat | Llama2-7B-chat |
|---|---|---|---|---|---|
| accuracy | **0.61** | 0.55 | 0.52 | 0.45 | 0.39 |

To further confirm the observations on the LegalBench-Instruct, (Colombo et al., 2024) conducted additional experiments on the legal tasks from the MMLU benchmark. The SaulLM-7B-Instruct model exhibited strong performance across all three tasks, including international law, professional law, and jurisprudence tasks.

Besides, Institute (2023) has not yet reported the specific classification accuracy of the BAI-Law-13B model, leaving its performance on certain benchmarks unclear.

Table 4: Comparison of LLMs on the MMLU benchmark.

| LLMs | SaulLM-7B-Instruct | Mistral-v1 | Mistral-v2 |
|---|---|---|---|
| International law | **0.69** | 0.62 | 0.65 |
| Professional law | **0.41** | 0.38 | 0.37 |
| Jurisprudence | **0.63** | 0.58 | 0.6 |

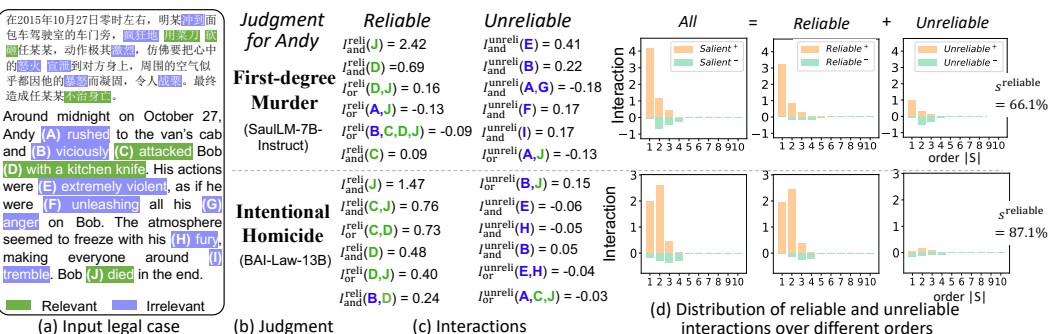

Figure 7: More results of judgments influenced by unreliable sentimental tokens. (a) A number of irrelevant tokens were annotated in the legal case, including unreliable sentimental tokens. Criminal actions were annotated as relevant tokens. We also translated the legal case to English as the input of the SaulLM-7B-Instruct model. (b) Judgements predicted by the two legal LLMs, which were both correct according to laws of the two countries. (c,d) We quantified the reliable and unreliable interaction effects of different orders. The SaulLM-7B-Instruct model used 66.1% reliable interaction effects, while the BAI-Law-13B model encoded 87.2% reliable interaction effects.

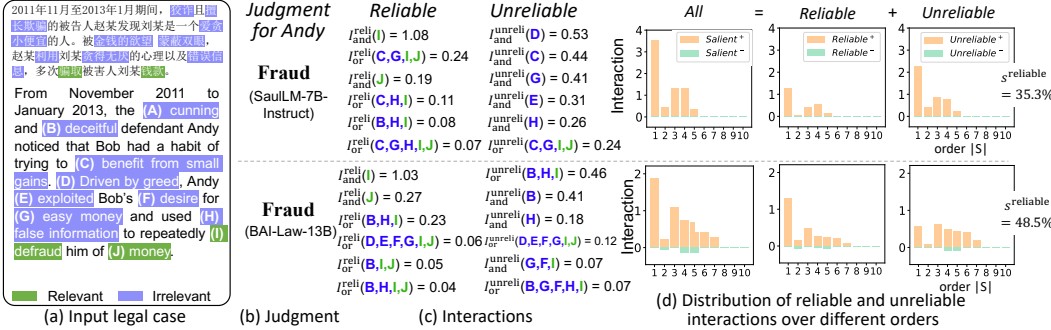

Figure 8: More results of judgments influenced by unreliable sentimental tokens. (d) The SaulLM-7B-Instruct model used 35.3% reliable interaction effects, while the BAI-Law-13B model encoded 48.5% reliable interaction effects.

# G  MORE EXPERIMENT RESULTS AND DETAILS

## G.1  MORE RESULTS OF JUDGMENTS INFLUENCED BY UNRELIABLE SENTIMENTAL TOKENS

We conducted more experiments to show the judgments influenced by unreliable sentimental tokens in Figure 7, Figure 8, and Figure 9, respectively. We observed that a considerable number of interactions contributing to the confidence score $v(\mathbf{x})$ were attributed to semantically irrelevant or unreliable sentimental tokens. In different legal cases, the ratio of reliable interaction effects to all salient interactions was within the range of 32.6% to 87.1%. It means that about 13~68% of interactions used semantically irrelevant tokens or unreliable sentimental tokens for the judgment.

## G.2  MORE RESULTS OF JUDGMENTS AFFECTED BY INCORRECT ENTITY MATCHING

We conducted more experiments to show the judgments affected by incorrect entity matching in Figure 10, Figure 11, and Figure 12, respectively. We observed that a considerable ratio of the confidence score $v(\mathbf{x})$ was mistakenly attributed to interactions on criminal actions made by incorrect

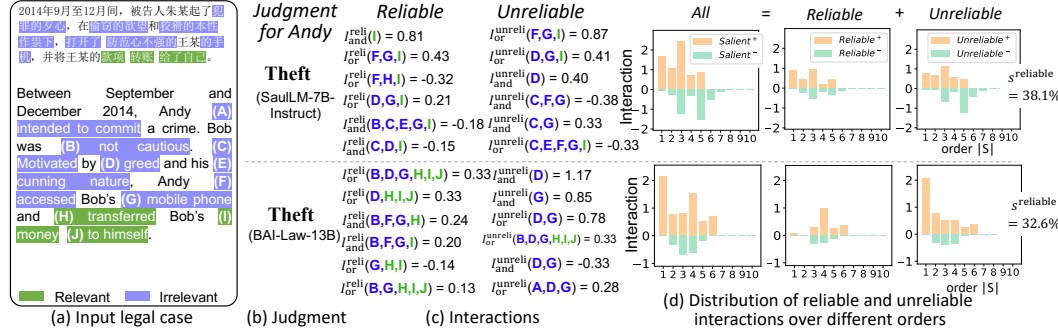

Figure 9: More results of judgments influenced by unreliable sentimental tokens. (d) The SaulLM-7B-Instruct model used 38.1% reliable interaction effects, while the BAI-Law-13B model encoded 32.6% reliable interaction effects.

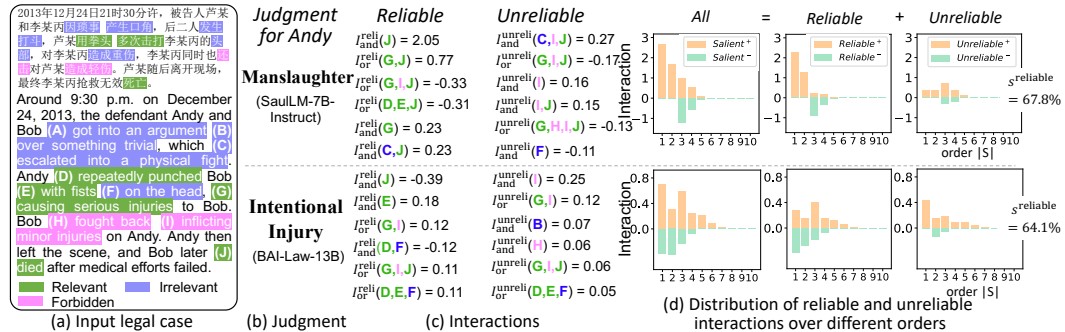

Figure 10: More results of judgments affected by incorrect entity matching. (a) A number of irrelevant tokens were annotated in the legal case, including the time and actions that were not the direct reason for the judgment. Criminal actions of the defendant were annotated as relevant tokens. Criminal actions of the unrelated person were annotated as forbidden tokens. (b) Judgements predicted by the two legal LLMs, which were both correct according to laws of the two countries. (c,d) We measured the reliable and unreliable interaction effects of different orders. The SaulLM-7B-Instruct model used 67.8% reliable interaction effects, while the BAI-Law-13B model encoded 64.1% reliable interaction effects.

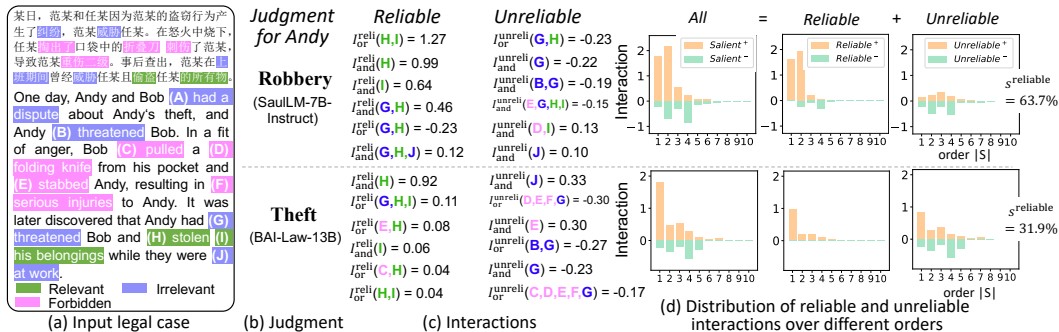

Figure 11: More results of judgments affected by incorrect entity matching. (d) The SaulLM-7B-Instruct model used 63.7% reliable interaction effects, while the BAI-Law-13B model encoded 31.9% reliable interaction effects.

entities. In different legal cases, the ratio of reliable interaction effects to all salient interactions was within the range of 31.9% to 67.8%. It means that about 22~68% of interactions used semantically irrelevant tokens for the judgment, or was mistakenly attributed on criminal actions made by incorrect entities.

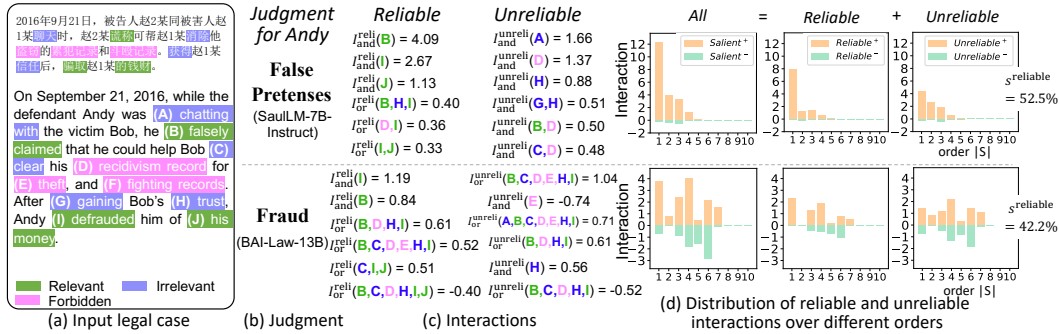

Figure 12: More results of judgments affected by incorrect entity matching. (d) The SaulLM-7B-Instruct model used 52.5% reliable interaction effects, while the BAI-Law-13B model encoded 42.2% reliable interaction effects.

### G.3 MORE RESULTS OF JUDGMENTS BIASED BY DISCRIMINATION IN OCCUPATION

**Experiment results of judgments biased by discrimination in occupation in Section 3.** Figure 16 illustrates additional examples of how occupation influences the judgment of the legal case, which were tested on the SaulLM-7B-Instruct model. It shows that if we replaced "*a judge*" with law-related occupations, such as "*a lawyer*" and "*a policeman*," the judgment remained "*robbery*." Besides, the occupation "*a programmer*" changed the judgment to "*not mentioned*." The interactions containing the occupation token (*i.e.*, "*a judge*", "*a lawyer*", "*a policeman*", "*a programmer*", and "*a volunteer*") were important factors that changed the ratio of reliable interactions from 81.4% to 84.0%. This suggested that the legal LLM sometimes had considerable occupation bias.

Futhermore, Figure 17 shows the test of the BAI-Law-13B model on the legal case, in which *Andy, the victim with varying occupations, was robbed of his belongings by two suspicious men*. Similarly, we found that the BAI-Law-13B model encoded interactions with the occupation tokens "*a judge*," which boosted the confidence of the judgment "*robbery*." More interestingly, if we substituted the occupation tokens "*a judge*" to "*a policeman*," the interaction of the occupation "*a policeman*," decreased from 0.29 to 0.11. The interactions containing the occupation token were important factors that changed the ratio of reliable interactions from 78.9% to 87.1%. This suggested that the legal LLM sometimes had considerable occupation bias.

**More results of judgments biased by discrimination in occupation.** We conducted more experiments to show the judgments biased by discrimination in occupation in Figure 13, Figure 14, and Figure 15, respectively. We found that the legal LLM usually used interactions on the occupation information to compute the confidence score $v(\mathbf{x})$. In different legal cases, the ratio of reliable interaction effects to all salient interactions was within the range of 30.1% to 63.7%. In particular, in Figure 13, changing the occupation from "*lawyer*" to "*programmer*" results in a decrease of the reliable interactions from 63.7% to 57.3%. The difference of interactions containing the occupation token changes the model output from "*Larceny*" to "*Theft*."

### G.4 EXPERIMENT DETAILS OF MASKED SAMPLES

This section discusses how to obtain the masked sample $\mathbf{x}_T, T \subseteq N$. Given the confidence score of a DNN $v(\mathbf{x})$ and an input sample $\mathbf{x} = [x_1, x_2, \cdots, x_n]^\intercal$ with $n$ input variables, if we arbitrarily mask the input sample $\mathbf{x}$, we can get $2^n$ different masked samples $\mathbf{x}_T, \forall T \subseteq N$. Specifically, for each input variable $i \in N \setminus T$, we replace it with the baseline value $b_i$ to represent its masked state. Let us use baseline values $\mathbf{b} = [b_1, b_2, \cdots, b_n]^\intercal$ to represent the masked state of a masked sample $\mathbf{x}_T$, *i.e.*,

$$(\mathbf{x}_T)_i = \begin{cases} x_i, & i \in T \\ b_i, & i \notin T \end{cases} \tag{18}$$

For sentences in a language generation task, the masking of input variables is performed at the embedding level. Following the approach of (Ren et al., 2024; Shen et al., 2023), we masked inputs at the embedding level by transforming sentence tokens into their corresponding embeddings. Given an input sentence $\mathbf{x} = [x_1, x_2, \cdots, x_n]^\intercal$ with $n$ input tokens, the $i$-th token $x_i$ is mapped to its

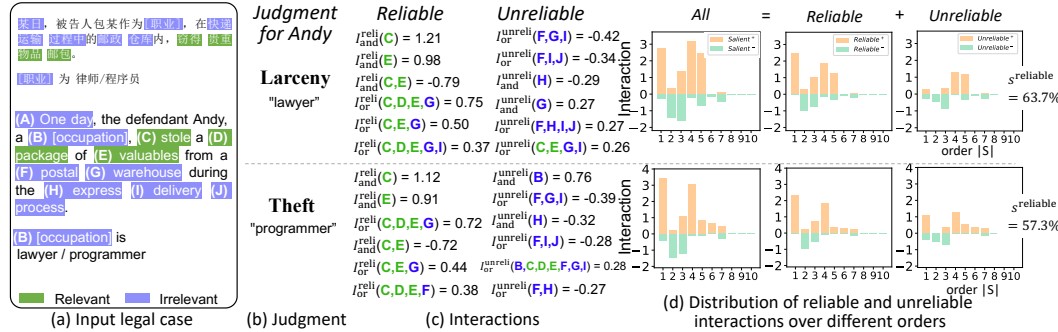

Figure 13: More results of judgments biased by discrimination in occupation. (a) A number of irrelevant tokens were annotated in the legal case, including the occupation, time and actions that are not the direct reason for the judgment. Criminal actions of the defendant were annotated as relevant tokens. (b) The SaulLM-7B-Instruct model predicted the judgment based on the legal case with different occupations, respectively. (c,d) We measured the reliable and unreliable interaction effects of different orders. When the occupation was set to "*lawyer*," the LLM used 63.7% reliable interaction effects. In comparison, when the occupation was set to "*programmer*," the LLM encoded 57.3% reliable interaction effects.

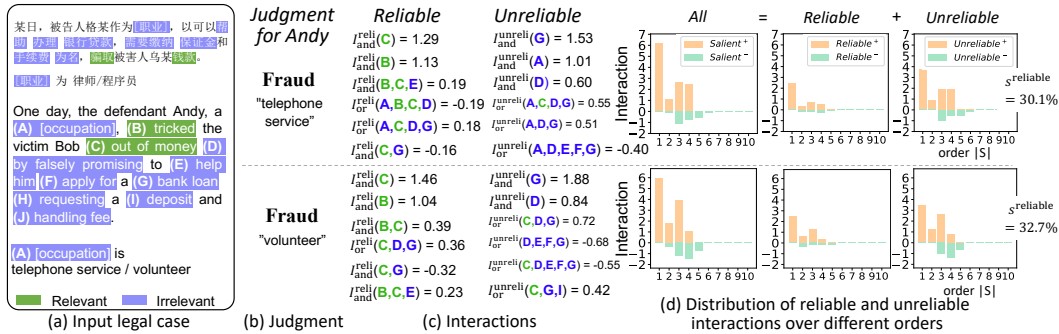

Figure 14: More results of judgments biased by discrimination in occupation. (b) The SaulLM-7B-Instruct model predicted the judgment based on the legal case with different occupations, respectively. (d) When the occupation was set to "*telephone service*," the LLM used 30.1% reliable interaction effects. In comparison, when the occupation was set to "*volunteer*," the LLM encoded 32.7% reliable interaction effects.

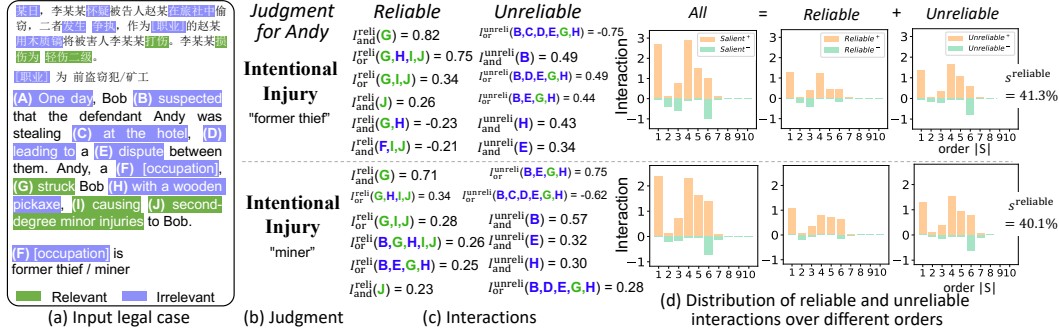

Figure 15: More results of judgments biased by discrimination in occupation. (b) The BAI-Law-13B model predicted the judgment based on the legal case with different occupations, respectively. (d) When the occupation was set to "*former thief*," the LLM used 41.3% reliable interaction effects. In comparison, when the occupation was set to "*miner*," the LLM encoded 40.1% reliable interaction effects.

embedding $e_i \in \mathbb{R}^d$, where $d$ is the dimension of the embedding layer. To obtain the masked sample $\mathbf{x}_T$, if $i \in N \setminus T$, the embedding is replaced with the (constant) baseline value $b_i \in \mathbb{R}^d$, *i.e.*, $e_i = b_i$.

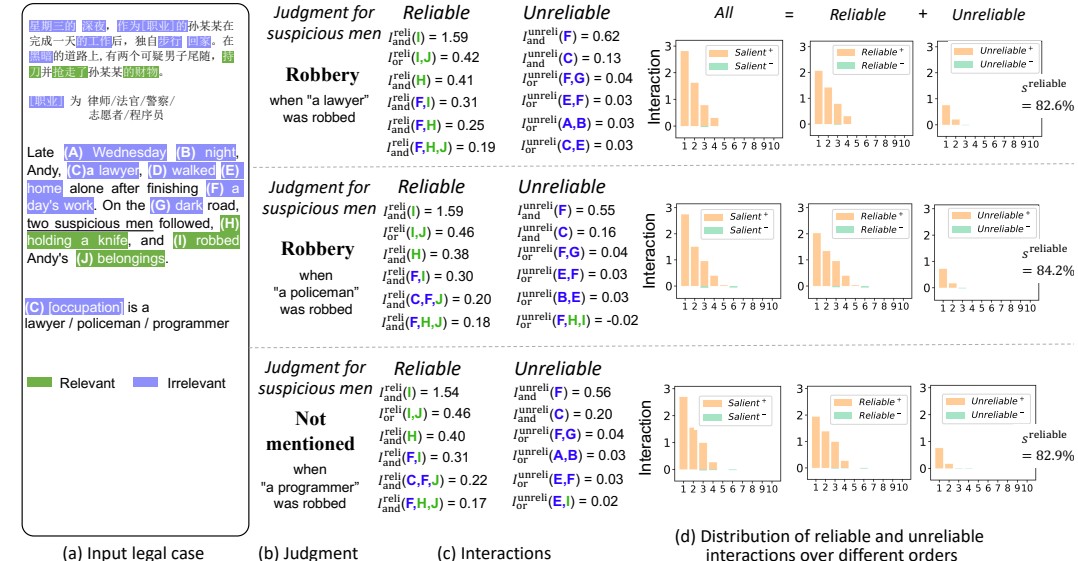

Figure 16: Visualization of judgments biased by discrimination in occupation. (a) A number of irrelevant tokens were annotated in the legal case, including the occupation, time and actions that are not the direct reason for the judgment. Criminal actions of the defendant were annotated as relevant tokens. (b) The SaulLM-7B-Instruct model predicted the judgment based on the legal case with different occupations, respectively. (c,d) We measured the reliable and unreliable interaction effects of different orders. When the occupation was set to "*a lawyer*," the LLM used 82.6% reliable interaction effects. In comparison, when the occupation was set to "*a policeman*," the LLM encoded 84.2% reliable interaction effects.

Otherwise, the embedding remains unchanged, *i.e.*, $e_i = e_i$. Following (Ren et al., 2023b), we trained the (constant) baseline value $b_i \in \mathbb{R}^d$ to extract the sparsest interactions.

## G.5 EXPERIMENT DETAILS FOR USING THE SAME DATASET FOR COMPARISON

This section presents the experiment details of using the CAIL2018 dataset (Xiao et al., 2018) to ensure a fair comparison between two legal LLMs. For the BAI-Law-13B model, a Chinese legal LLM, we directly analyzed the Chinese legal cases from the CAIL2018 dataset. In contrast, for the SaulLM-7B-Instruct model, an English legal LLM, we translated the Chinese legal cases into English and performed the analysis on the translated cases, to enable fair comparisons. To simplify the explanation and avoid ambiguity, we only explained the decision-making logic on legal cases, which were correctly judged by the LLM.

Starting with a complete fact descriptions of the legal case from the CAIL2018 dataset, we first condensed the case by removing descriptive details irrelevant to the judgment, retaining only the most informative tokens, such as the time, location, people, and events. To prompt the model to deliver its judgment, we added a structured prompt designed to extract a concise answer. The format is as follows:

"*Question: [Fact descriptions of the case]. What crime did [the defendant] commit? Briefly answer the specific charge in one word. Answer: The specific charge is*"

Here, *[Fact descriptions of the case]* is replaced with the details of the specific legal case, and *[the defendant]* is substituted with the name of the defendant.

To identify potential representation flaws behind the seemingly correct language generation results of legal LLMs, we introduced special tokens that were irrelevant to the judgments. For cases to assess if judgments were influenced by unreliable sentimental tokens, we added such tokens to describe actions in the legal case. We then observed whether a substantial portion of the interactions contributing to the confidence score $v(\mathbf{x})$ were associated with semantically irrelevant or unreliable sentimental tokens. Similarly, in cases where we aimed to detect potential bias based on occupation, we included irrelevant occupation-related tokens for the defendants or victims, and analyzed whether

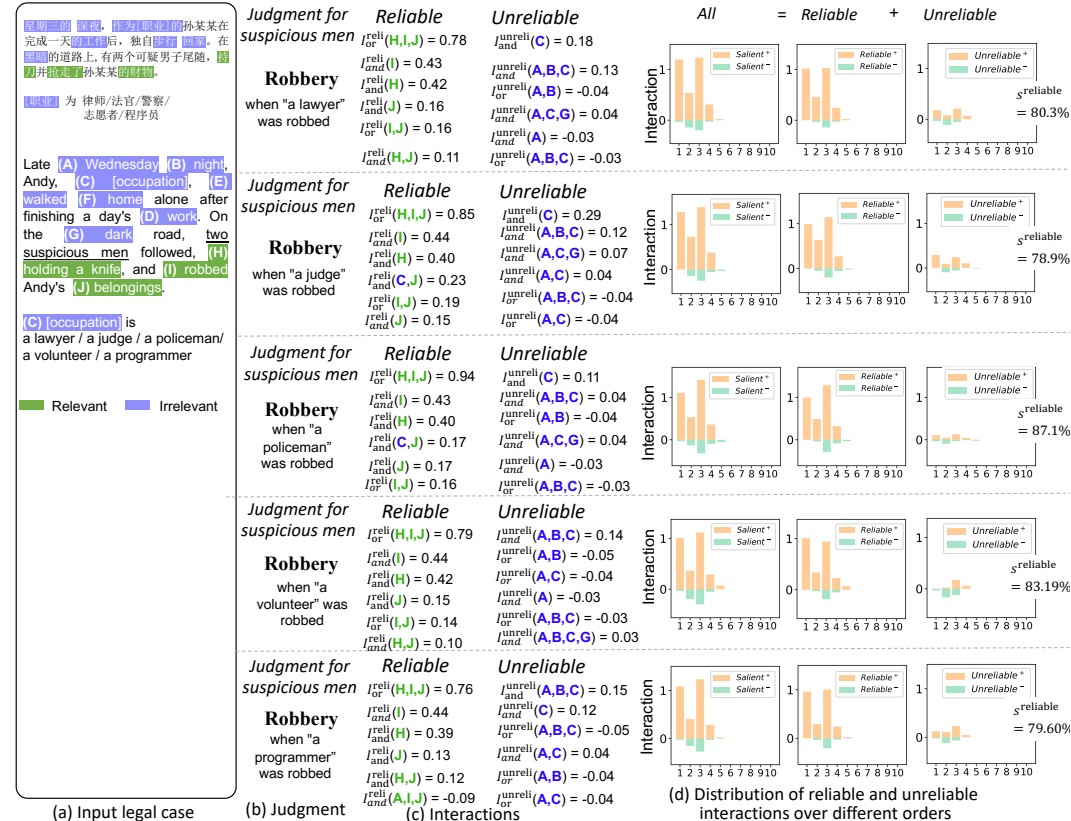

Figure 17: Visualization of judgments biased by discrimination in occupation. (a) A number of irrelevant tokens were annotated in the legal case, including the occupation, time and actions that are not the direct reason for the judgment. Criminal actions of the defendant were annotated as relevant tokens. (b) The BAI-Law-13B model predicted the judgment based on the legal case with different occupations, respectively. (c,d) We measured the reliable and unreliable interaction effects of different orders. When the occupation was set to "*a judge*," the LLM used 78.9% reliable interaction effects. In comparison, when the occupation was set to "*a policeman*," the LLM encoded 87.1% reliable interaction effects.

the legal LLM leveraged these occupation-related tokens to compute the confidence score $v(\mathbf{x})$ in Equation (1).

Finally, we show the selection of input variables for extracting interactions. As discussed in Section 2.1, given an input sample $\mathbf{x}$ with $n$ input variables, we can extracted at most $2^{n+1}$ AND-OR interactions to compute the confidence score $v(\mathbf{x})$. Consequently, the computational cost for extracting interactions increases exponentially with the number of input variables. To alleviate this issue, we followed (Ren et al., 2024; Shen et al., 2023) to select a set of tokens as input variables, while keeping the remaining tokens as a constant background in Appendix G.4, to compute interactions among the selected variables. Specifically, we selected 10 informative input variables (tokens or phrases) for each legal case. These input variables were manually selected based on their informativeness for judgements. It was ensured that the removal of all input variables would substantially change the legal judgment result.

