# OpenReview forum: "Alignment Between the Decision-Making Logic of LLMs and Human Cognition: A Case Study on Legal LLMs"
_ICLR.cc/2025/Conference — ICLR 2025 Conference Withdrawn Submission_

### Official Review · Reviewer_gchB · 2024-10-27

**Soundness:** 2
**Presentation:** 3
**Contribution:** 2
**Rating:** 5
**Confidence:** 3

**Summary:**

The paper investigates the alignment between the decision-making logic of Large Language Models (LLMs) and human cognition, with a focus on legal applications. It introduces a novel approach to evaluate the internal decision-making logic behind LLMs' outputs, addressing a critical gap in traditional evaluation methods that only assess output correctness. The study emphasizes the importance of aligning LLM's internal logic with human reasoning to build trust, especially in high-stakes scenarios like legal judgments. The paper proposes metrics to differentiate reliable and unreliable interaction effects within the LLMs, providing a structured way to assess and improve the models' alignment with human cognitive processes.

**Strengths:**

1. Innovative methodology: The paper proposes a novel framework to assess the correctness of the decision-making logic in LLMs, moving beyond surface-level output evaluation. This approach is particularly pertinent as it addresses the trust issues associated with LLM deployment in sensitive areas.

2. Insightful empirical findings: The study reveals significant discrepancies in how LLMs process information compared to human reasoning, with detailed case studies illustrating the potential risks in legal applications.

3. Theoretical depth: The research is grounded in thorough theoretical analysis, supported by recent studies in explainable AI. It leverages interaction-based explanations to quantitatively measure the alignment between LLM outputs and human decision-making logic.

4. Comprehensive literature review: The paper effectively situates its contributions within the broader context of AI trustworthiness and cognitive alignment, synthesizing insights from various sub-fields of AI and cognitive science.

**Weaknesses:**

1. Motivation and Conceptual Framework (Sections 1): The motivation behind focusing on the alignment of decision-making logic with human cognition is clear but could be expanded to directly link to specific impacts in legal applications. For instance, the paper could include examples or case studies where misalignment has led to practical errors in judgment predictions by LLMs, reinforcing the need for the proposed methodology.

2. Application of Findings (Sections 1): The paper should detail how understanding reliable and unreliable interaction effects can directly enhance the performance and safety of LLMs. This could include suggestions for practical implementations, such as modifications to training procedures, model architecture adjustments, or data preprocessing techniques that could help minimize unreliable interactions.

3. Section 2: The methodology for implementing the proposed metrics is currently outlined at a high level, which might be challenging for practitioners without a deep background in AI. The paper could benefit from a step-by-step guide or a pseudocode representation of the methodology, along with examples of how to compute reliable and unreliable interaction effects.

4. Experiment Design (Section 3): The current experiments predominantly focus on legal LLMs. To enhance the robustness of the findings, the authors could include additional LLM architectures used in other domains (e.g., healthcare, finance) to see if the findings hold consistently across different settings. This expansion would help validate the universality of the proposed metrics and their applicability to a broader range of practical applications.

5. Experimentation (Section 3): While the experiments demonstrate discrepancies in decision-making logic, the paper could strengthen its claims by statistically analyzing the impact of these discrepancies on the performance of LLMs in real-world scenarios. This might involve quantitative measures of how misalignments affect the outcomes and comparing them across different models or settings.

**Questions:**

To enhance the robustness and applicability of the paper, authors should consider expanding the experimental scope by incorporating additional LLM architectures and domains such as healthcare or finance. This expansion would help validate the universality of the proposed metrics. Detailed implementation guidance, including pseudocode or example code, would also be beneficial, particularly for practitioners with limited AI expertise. Questions regarding the statistical analysis of results and the generalizability of the findings across different datasets and models are crucial. Addressing these aspects could strengthen the paper’s claims and ensure the findings are applicable in real-world settings, significantly enhancing the paper's impact and utility across various fields.

---

### Official Review · Reviewer_Uug2 · 2024-10-29

**Soundness:** 2
**Presentation:** 1
**Contribution:** 1
**Rating:** 3
**Confidence:** 3

**Summary:**

The paper attempts to develop models to understand the behaviour of LLM reasoning in legal applications. The overall conclusion is, not surprisingly, that LLMs are frequently buggy.

The rationale for the paper seems somewhat clear, but I found the implementation to be a mess. I was very concerned about the presentation in the paper, which is poor, but also the primitive decision making logic seems to demonstrate very little awareness of actualy legal reasoning, or more worryingly, the huge literature on legal reasoning in AI. There are MANY models of legal reasoning and argumentation in AI: the paper literally doesn't cite any of that.

Overall, my sense is that the authors have some technically interesting ideas, but I find the weaknesses in the work too substantial to merit publication at ICLR.

**Strengths:**

The paper addresses an important topic: can LLMs do legal reasoning? What models might correcrtly describe their behaviour? Some work is done on developing a model to account for these, and some experiments are done to evaluate them.

**Weaknesses:**

The presentation is very poor, to the point where I found it really quite hard to continue reading.

The ideas presented seem technically OK, but the AND/OR logic is superficial.

The paper doesn't demonstrate any awareness of the enormous literature on AI and law. See eg...

http://www.iaail.org
https://www.lawsociety.org.uk/topics/research/ai-artificial-intelligence-and-the-legal-profession

It is quite hard to take the work seriously as a study on AI and law given this.

**Questions:**

What is the model that correctly characterises the outputs an LLM produces in legal reasoning settings?

Can you give a precise logical characterisation of the reasoning processes you are capturing?

---

### Official Review · Reviewer_XEP3 · 2024-11-04

**Soundness:** 2
**Presentation:** 1
**Contribution:** 2
**Rating:** 3
**Confidence:** 3

**Summary:**

This paper studies the alignment between the decision-making logic of LLMs and human cognition in the domain of legal LLMs. It proposes a method to evaluate the correctness of LLMs' decision-making logic and designs metrics to quantify different kinds of interaction effects. Through evaluations on the CAIL 2018 dataset using two legal LLMs, the authors find that LLMs often reply on problematic interactions to make judgments, even though the LLMs achieve high accuracy in judgment prediction. The findings are suggestive of current limitations of LLMs.

**Strengths:**

- This works engages with important topics: legal applications of LLMs, alignment, and explainable AI. It reinforces the fact that accuracy alone cannot guarantee the applicability, reliability, and trustworthiness of LLMs.

- The framework of And-Or interactions and the measure of interaction effects are technically interesting and seemingly useful. They help identify three specific types of problems that the LLMs exhibit.

- The methods proposed here can potentially be applied to other domains and problems.

**Weaknesses:**

- A major weakness of this work is its presentation—it is not clearly written and not in a professionally acceptable style. Substantial rewriting might be needed to improve it to publication standards.
  - It is essential for any paper to include "related work" in the main text (not just in the appendix). It is important for readers to see what are related areas of research and how this work differs from them easily and straightforwardly. Also note that reviewers are not required to read the appendix, another reason why the authors should place all of the most important parts of a paper in the main text.
  - I think Figures 3-5 are supposed to be the main result figures of the paper, but they all seem to be about specific examples. It is good to have some of those, but it is also important to show dataset-level aggregate results (which the authors include in the appendix).
- Another major weakness is about the term "human cognition". This term appears in the title, and it is supposed to be an integral part of the paper, appearing many times. However, it is not defined in the paper, and no citations to it is given. What do the authors mean by "human cognition"? Typically, when a paper makes claims about human cognition, it needs to cite or include studies/data/models/theories that engage with cognitive science. Does the term in this paper just mean "human judgments" in prior work or the "common sense" of the authors? That would be fine, but would need to be explicitly stated. The phrase "according to human cognition" is not informative.
- Here are some specific comments and suggestions:
  - I recommend that the authors motivate a bit more why they "focus on the legal LLM as a case study" (Line 34). Trustworthiness, safety, and the decision mechanisms of LLMs are all general topics that are relevant to many domains. The legal domain is obviously important and high impact, but a reader might wonder why the authors choose it in this context.
  - In the introduction of the paper, some of the sentences are not very appropriate or do not sound very natural. For example, in Lines 37-38, can you support the claim "The alignment of internal logic via communication is the reason why people naturally trust each other"? Moreover, the word "Typically" in Line 44 feels a bit out of place.
  - The first sentence of Section 2, "Although there is no widely-accepted definition of concepts..." does not seem necessary. It is true that no one definition/theory is viewed as correct, but some are much more widely accepted than others (e.g., definitional/prototype/exemplar/"theory-" theories are all very influential). It does not look like that these theoretical issues really matter here for the purpose of this paper.
  - I would encourage the authors to discuss and interpret their results and findings more towards the end of the paper. For example, how might one mitigate the problems identified here? Should we expect bigger models to be better w.r.t the problems? What are some implications of the problems?

**Questions:**

See the "Weaknesses" section.

---

### Official Review · Reviewer_uKsd · 2024-11-04

**Soundness:** 3
**Presentation:** 2
**Contribution:** 2
**Rating:** 5
**Confidence:** 3

**Summary:**

In this paper, the authors analyzed the interactions that impact a legal LLM's confidence score of the true judgment and assessed their alignment with human cognition of the case.
To this end, the authors group all input tokens involved in the interactions into three types,

i.e., the relevant, irrelevant, and forbidden
tokens,  based on their ground truth significance to the decision.

**Strengths:**

1. A new metrics based on LLM interactions with legal cases
2. Evaluation of the alignment between the LLM’s logic and human cognition

**Weaknesses:**

Section 2.2 "Specifically, the set of all input variables N is partitioned into three mutually disjoint subsets, i.e.,
the set of relevant tokens R, the set of irrelevant tokens I, and the set of forbidden tokens F, subject
to R ∪ I ∪ F = N, with R ∩ I = ∅, R ∩ F = ∅, and I ∩ F = ∅, according to human cognition."
How are these tokens partitioned? Were there any inter-rater agreements? Was done by crowdsourced? What are the guidelines for partitioning?

**Questions:**

Please refer to weakness

---

### Official Review · Reviewer_vBxd · 2024-11-04

**Soundness:** 2
**Presentation:** 1
**Contribution:** 2
**Rating:** 3
**Confidence:** 3

**Summary:**

Paper investigates the relationship between decision-making logic of legal LLMs and human cognition. The paper uses interaction-based explanations to analyse the decision-making process of LLMs. Further, the paper categorises input tokes into three categories, those relevant for the judgment, those irrelevant for the judgment and forbidden tokens (those common in legal cases, but may lead to incorrect logic). Based on these categories, the paper distinguishes between reliable interaction effects (aligning with human cognition) and unreliable interaction effects (not matching human cognition).

**Strengths:**

I like that the paper shifts the focus from evaluating the correctness of the final output to examining the internal reasoning process of LLMs.

**Weaknesses:**

Related works should be in the main body of the paper, not in the appendix and the current work should explicitly related to the relevant previous work.

I don't really understand how the tokens were categorised 'according to human cognition'. For example, authors claim that 'Relevant tokens refer to tokens that are closely related to or serve as the direct reason for the judgment, according to human cognition'. But these are just relevant pieces of information, that people may even miss. So I don't really understand what is mean by human condition here. For forbidden tokens authors claim that these may lead to incorrect judgments. But it's not clear why. What relevant data from human decision-making have authors consulted here?

**Questions:**

What relevant data from human decision-making have authors consulted here? How the three categories relate to human cognition?

---

### Official Review · Reviewer_mLnC · 2024-11-04

**Soundness:** 1
**Presentation:** 1
**Contribution:** 2
**Rating:** 3
**Confidence:** 3

**Summary:**

The authors have used AND-OR interaction explanations to assess whether tokens used for coming up with decisions by legal LLMs aligh with tokens that a human decision-maker would use in their stead.  The evaluation is carried out on the CAIL18 data set with two legal LLMs.

**Strengths:**

**Originality**: To the best of my understanding this is an application paper. In short, it applies existing algorithms for extracting AND-OR interactions -- which, roughly speaking, are a kind of local explanation -- to understand whether legal LLMs make use of the same tokens that a human would in their decision process.  The analysis and the experimental setup themselves appear to be novel.

**Quality**: At a high level, the experimental setup is sensible, but please see the Weaknesses.

**Significance**: The results are potentially of interest to the legal LLM community.

**Weaknesses:**

**Originality**: It is a difficult to distinguish what is entirely new and what is instead taken from previous work.  For instance, it is not clear to me whether the proofs in Appendices B and C are new.  This is because the authors keep repeating how much work AND-OR interactions have received recently (which is fine) but do not clarify in sufficient detail what is really new compared to previous research in their treatment of AND-OR interactions.

- **Suggestion**: please explicitly state which aspects of the AND-OR interaction analysis are novel contributions of this paper versus building on prior work.

**Clarity**: I found the text unnecessarily difficult to read.  This stems from several issues, both linguistic (there are several linguistic mistakes and idiosyncrasies throughout the text, I have highlighted a few instances below) and structural (often, intuition for terms and equations is provided too late).  Nomenclature choices also complicate matters (I am still confused about the distinction between reliable
and unreliable interactions: they are (un)reliable for doing what?).  I find Figure 1 is more complex than it needs be, hindering intuition.
Unfortunately it is a bit difficult to pinpoint a specific paragraph or section that needs more work, the issue is quite diffuse.

**Quality**: The key point that the paper failed to address, for me, is that there appears to be a disconnect between the theoretical results and the claim that the paper analyzes the "decision-making logic" of the LLM.  The former guarantee that the explanation is faithful: it perfectly recovers the model's outputs for any masked input.  So far so good.  The latter, however, is about what the model internally thinks. I don't see a strong link between these two aspects.  More concretely: to the best of my understanding the theoretical results do not say anything about AND-OR explanations being *unique*. Now, if there exist two distinct AND-OR explanations that both faithfully explain the same model's decision, and they are incompatible, we can ask: which one of the two does really capture the LLM's reasoning process?  I don't think faithfulness guarantees translate into a guarantee that AND-OR explanations capture the model's actual reasoning process.  They definitely capture its decision surface, I'm not sure they also capture its "decision-making logic".  The analogy that comes to mind is that between post-hoc explainers like LIME and SHAP (which focus on the decision surface) and mechanistic interpretability (which looks at the inner mechanisms of models).  The text leads me to believe that the authors may be confusing these two aspects.  Of course I could be wrong, and I would appreciate a clarification.

- **Suggestion**: please directly address the potential disconnect between faithfully explaining the model's outputs and capturing its internal reasoning process. Specifically, you could discuss any limitations of using AND-OR interactions to infer the LLM's actual decision-making logic, rather than just its input-output behavior.  I also suspect that you don't need to claim to be studying the "decision-making logic" of the LLM to make sense of (or apply) the results of your experiment, so another option would be to less the emphasis on decision-making logic altogether.

Another major issue is that it is not clear how informative tokens were selected for evaluation.  This step is crucial, yet it is not described, possibly due to time constraints?  If left unresolved, this is a blocker for me.

- **Suggestion**: provide a detailed description of the token selection process, including any criteria used to determine which tokens were considered "informative" for the legal judgments, ideally in the methodology section.

**Significance**: Considering all existing literature on shortcut learning (e.g., [1, 2], but there are is so much more), it is not too surprising that legal LLMs are prone to use tokens that are not aligned to those that a human decision maker would use.  This limits potential impact.

[1] Ye, W., Zheng, G., Cao, X., Ma, Y., & Zhang, A. (2024). Spurious correlations in machine learning: A survey. arXiv preprint arXiv:2402.12715.
[2] Teso, S., Alkan, Ö., Stammer, W., & Daly, E. (2023). Leveraging explanations in interactive machine learning: An overview. Frontiers in Artificial Intelligence, 6, 1066049.

Minor Issues
----

- p 1: "The alignment of internal logic via communication is the reason why people naturally trust each other. Particularly, in high-stakes tasks such as autonomous driving (Grigorescu et al., 2020), the lack of alignment between AI models and human users makes people would rather delegate work to humans and tolerate potential errors, than trust highly accurate AI models."  This claim has no backing.  Please provide references in support.

- p 1: "propoerty".

- line 73: "which have not" -> which has not.

- line 125: "an LLM usually encodes": what do you mean by "usually"?  Does this
  entail that the decomposition into AND-OR interactions does not always apply?
  This seems inconsistent with the following claim that "These salient interactions are taken as the AND-OR logic really encoded by the LLM." (line 143).  I'd appreciate
  a clarification, also in light of the universal matching property.  The same
  comment applies to line 185.

- line 140: "sparest".

- line 204: given that N is a set, it should be calligraphic, just like calR,
  calI, calF.

- line 212: the input variables are binary, right?  It would make sense to
  mention this, for clarity.

- line 231: "are naturally all represent crucial facts".

- line 250: "SaulLM-7B-Instruct was" and following: was -> is.

- line 103: I don't quite understand why the text mentions "concepts", provided
  that concepts are not well defined and they don't seem to play a central
  role in the paper.  I suggest the authors to avoid mentioning concepts
  altogether: this discussion does not seem to add anything to the message.
  (It's not I don't like concepts -- it is quite the contrary -- they just feel
  irrelevant to this work: all that matter is that AND-OR interactions
  represent a sensible way of approximating the LLM's reasoning process, at
  least for a fixed input x.)

- line 307: "order(S) = |S|": this shorthand is unnecessary.

**Questions:**

I would appreciate if the authors could provide insights about all the major issues I pointed out above.

---

### Note · Authors · 2024-11-22

I have read and agree with the venue's withdrawal policy on behalf of myself and my co-authors.